# Scalable Flood Level Trend Monitoring with Surveillance Cameras using a Deep Convolutional Neural Network

Matthew Moy de Vitry [1,2], Simon Kramer [2], Jan Dirk Wegner [3], and João P. Leitão[1]

[1] Department of Urban Water Management, Eawag - Swiss Federal Institute of Aquatic Science and Technology, 8600 Dübendorf, Switzerland

[2] Institute of Environmental Engineering, ETH Zurich, 8093 Zürich, Switzerland

[3] EcoVision Lab, Photogrammetry and Remote Sensing group, ETH Zurich, 8093 Zürich, Switzerland

*Correspondence to*: Matthew Moy de Vitry (matthew.moydevitry@eawag.ch)

**Abstract.** In many countries, urban flooding due to local, intense rainfall is expected to become more frequent because of climate change and urbanization. Cities trying to adapt to this growing risk are challenged by a chronic lack of surface flooding data that is needed for flood risk assessment and planning. In this work, we propose a new approach that exploits existing surveillance camera systems to provide qualitative flood level trend information at scale. The approach uses a deep convolutional neural network (DCNN) to detect floodwater in surveillance footage and a novel qualitative flood index (SOFI) as a proxy for water level fluctuations visible from a surveillance camera's viewpoint. To demonstrate the approach, we trained the DCNN on 1218 flooding images collected from the Internet and applied it to six surveillance videos representing different flooding and lighting conditions. The SOFI signal obtained from the videos had on average a 75% correlation to the actual water level fluctuation. By retraining the DCNN with a few frames from a given video, correlation is increased to 85% on average. The results confirm that the approach is versatile, with the potential to be applied to a variety of surveillance camera models and flooding situations without the need for on-site camera calibration. Thanks to this flexibility, this approach could be a cheap and highly scalable alternative to conventional sensing methods.

## 1 Introduction

### 1.1 The need for urban pluvial flood monitoring data

Urban pluvial floods are floods caused by intense local rainfall in urban catchments, where drainage systems are usually not designed to cope with storm events of more than a 10-year return period. Although the full impact of such flood events is difficult to gauge because of reporting and knowledge gaps (Paprotny et al., 2018; van Riel, 2011), some studies estimate the societal cost of small but frequent urban pluvial floods to be comparable to the cost of large, infrequent fluvial flooding events (Jiang et al., 2018b; ten Veldhuis, 2011). Additionally, it is generally acknowledged that the frequency of urban pluvial floods will increase under the driving forces of climate change and urbanization (Skougaard Kaspersen et al., 2017; Zahnt et al., 2018).

To cope with urban pluvial flood risk, urban drainage managers must understand long-term flooding trends, design appropriate flood mitigation solutions in the medium term, and provide flood alerts in the short term. Numerical flood modelling is a widely used tool for all these tasks, but a certain amount of data is needed for modelling. Data pertaining to drainage infrastructure, land use, and elevation is required to construct a model, and rainfall data is required to test the model on past rain events. Additionally, flood monitoring data allows for model calibration, which is essential for improving the accuracy of urban drainage models (Tscheikner-Gratl et al., 2016). However, conventional sensors are ill suited to urban environments, where vehicles can disturb the flow and vandalism is a high risk. Similarly, remote sensing is not able to provide data with sufficient spatiotemporal resolution. The lack of monitoring methods and ensuing data scarcity are frequently decried in the urban pluvial flood modelling community (Gaitan et al., 2016; Hunter et al., 2008; El Kadi Abderrezzak et al., 2009; Leandro et al., 2009). In this context, researchers and practitioners have turned to alternative sources of data such as surveillance footage (Liu et al., 2015; Lv et al., 2018), ultrasonic-infrared sensor combinations (Mousa et al., 2016), field surveys (Kim et al., 2014), social media and apps (Wang et al., 2018),  and first-hand reports (Kim et al., 2014; Yu et al., 2016).

Although quantitative information (e.g. water level) is commonly sought for, studies show that even qualitative data are useful for calibrating hydraulic and hydrological models. Van Meerveld et al. (2017) calibrated a bucket-type hydrological model with stream level data transformed into sparse, dimensionless class information, such as might be collected by citizen scientists. To accomplish this, the authors used a genetic algorithm (Seibert, 2000) for parameter estimation and the Spearman rank correlation coefficient as an objective function. In doing so, they demonstrated that information could be gained from water level trend data with a conventional calibration toolset. Exploiting a different kind of qualitative information, Wani et al. (2017) showed that binary data from a combined sewer overflow to estimate the parameter value distributions of an urban drainage model. In their research, the Bayesian framework used allowed the reduction of parameter and model uncertainty to be described explicitly. We therefore also expect unconventional, qualitative monitoring data to be valuable for the estimation of parameters in urban pluvial flood models.

## 1.2 Surveillance cameras as a data source

Surveillance footage has several advantages when used for flood monitoring. First, many municipalities have already invested in a network of surveillance cameras. In the cities investigated by Goold et al. (2010), these networks usually have fifty to several hundred cameras. In certain cities, however, camera systems operated by institutions are also integrated in the municipal monitoring network. For example, the Command and Control Centre for the London Police is reported to have access to 60'000 cameras (Goold et al., 2010), and the police in Paris have access to 10'000 cameras operated by partners (Sperber et al., 2013). The second advantage of surveillance cameras is their high reliability, since their utility for traffic surveillance and crime reduction depends on continuous operation.

However, the use of surveillance footage for flood monitoring has complications. First, camera placement is generally controlled by outside parties for security purposes, thus critical flooding locations may be only partially visible in the footage

or even completely missed. Second, the personal privacy of individuals visible in the footage must be protected. Finally, the interpretation of surveillance footage into a signal that can be assimilated into a flood model is not trivial.

## 1.3 Automatic water level monitoring with surveillance images

While manual reading of water level from surveillance images is possible (e.g. in the study of Liu et al. (2015)), it is both prohibitively labor-intensive at scale and potentially critical from a privacy perspective. Automatic image analysis helps overcome these hurdles, and has already been the subject of research. The following publications provide the current state of the art of automatic water level estimation from ground-level images.

In the work of Lo et al. (2015), video frames are segmented into a number of visually distinct areas using a graph-based approach. The area corresponding to water is identified thanks to an operator-provided "seed", and the water level is qualitatively assessed by comparing the water area to virtual markers placed in the image by the operator. With a more camera-specific solution, Sakaino (2016) estimates water levels with a supervised histogram-based approach which assumes a straight water line on a wall visible in the footage. Similarly, Kim et al. (2011) used a ruler in the camera's field of view as a reference for the water level measurement. A similar approach is used by Bhola et al. (2018), who used the size of large objects like bridges to estimate the real height of automatically detected water surface in the image. Although these methods work well, they rely on in-situ measurements and site-specific calibration, and may be challenging to apply to a large number of cameras.

A more modern approach for image-based flood level estimation has been proposed by Jiang et al. (2018a). The authors use a deep convolutional neural network to extract image features and then apply a regression to infer water level. Although the results are positive, the approach requires that the neural network and regression be retrained for each camera. Thus, the method is probably most valuable for providing redundancy to existing water level readings and not as a scalable flood monitoring solution. Recently, a new approach has been proposed, which theoretically overcomes this problem by estimating water depth from the immersion of ubiquitous reference objects (e.g. bicycles) of known height (Chaudhary et al., 2019; Jiang et al., 2019). However, this approach requires that such objects be visible in the scene in order to provide information.

## 1.4 Objective of the present work

In this work, we propose a novel and highly scalable approach to automatically extract local flood level fluctuations from surveillance footage. By proposing this approach, we aim to provide a tool that can exploit existing surveillance infrastructure to furnish much-needed flood information to urban flood modelers and decision-makers. By making scalability a priority, we hope to facilitate adoption of the tool by practitioners, especially in cities where extensive surveillance camera systems are already in place.

## 2 Materials and methods

Our approach consists in a two-step processing pipeline that combines automatic image analysis with data aggregation (Fig. 1). In a first step, floodwater is segmented in individual video frames with a deep convolutional network (DCNN). The segmented frames are then summarized with an index (SOFI) that qualifies the visible extent and thereby the depth of the water over time. We evaluate the performance of this approach using footage from surveillance cameras during various flood events. Additionally, we investigate how the data used to train the DCNN influences both segmentation performance and the information content of the SOFI.

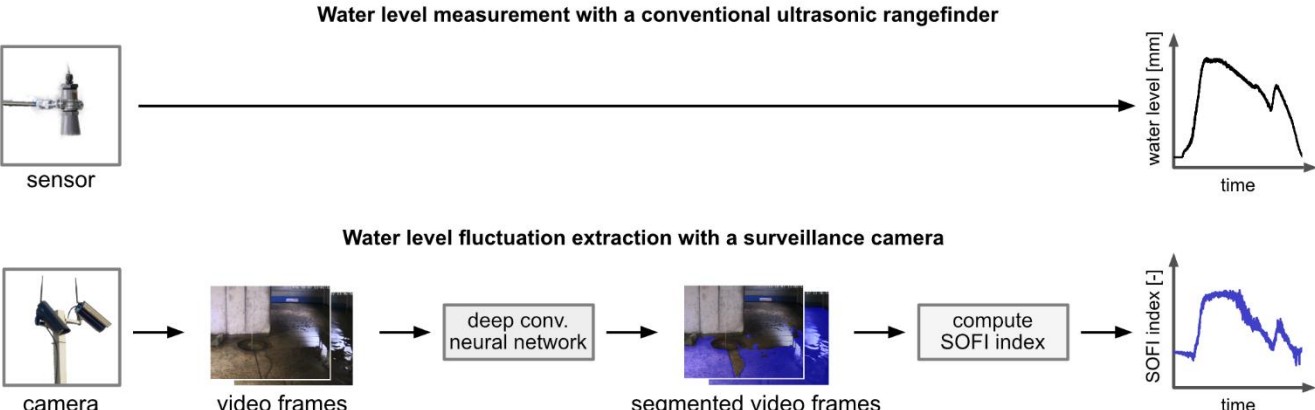

**Figure 1: As an alternative to conventional sensors (top), flood trend information is extracted from surveillance footage by computing the fractional water-covered area (SOFI) over a series of video frames (bottom).**

### 2.1 Flood water segmentation

#### 2.1.1 Image segmentation with deep convolutional neural networks

Semantic segmentation is the task of annotating each pixel in an image according to a predefined taxonomy. The most recent advances in image segmentation have been made with DCNNs (He et al., 2017), so it is of value to apply this powerful tool to the problem of flood segmentation. DCNNs are a subset of artificial neural networks (ANN), machine learning models with a structure that mimics the structure of neurons in the brain. In the case of DCNNs, images are interpreted through consecutive convolutional (matrix-like) layers that extract and combine information at varying levels of abstraction.

Although the concept of DCNNs originated in the 80s (Fukushima, 1980), their success for non-trivial problems requires large training sets and computational resources that have only become available relatively recently. An important breakthrough in DCNN development was the fully convoluted network (FCN) introduced by Long et al. (2015), in which the fully connected layers responsible for generating class labels are also formulated as convolutional layers, thereby providing spatially explicit label maps. However, FCN suffered an issue of resolution loss. To solve this issue, Noh et al. (2015) combined FCN with a "deconvolution network", a network that predates FCN (Zeiler et al., 2011) and consists in upsampling and unpooling layers.

## 2.1.2 Water segmentation with U-net

The DCNN architecture used for water segmentation in this work is that of U-net (Ronneberger et al., 2015). U-net builds on the FCN architecture, but differs in that the decoding layers have as many features as their respective encoding layers, which allows the network to propagate context and texture information to the final layers. Additionally, U-net implements "skip" connections to preserve details and object boundaries, by carrying information directly from the encoding to the decoding layers. The U-net architecture is well-suited to the water segmentation problem because of its relatively compact size compared to more recent state-of-the-art semantic segmentation architectures, such as Mask R-CNN (He et al., 2017). The smaller size makes it both easier to train with small datasets (like the one available for this study) and faster to run, which is useful for flood monitoring. To code the DCNN, we built on an open source implementation of U-net (Pröve, 2017) that uses Keras (Chollet and others, 2015) to interface with the TensorFlow library (Abadi et al., 2016).

After exploring a range of hyperparameter values (layer depth, feature size, etc.), we found the following network structure (Fig. 2) to have the best combination of performance and generalization potential for the flood segmentation problem. As input, the network takes color images with a resolution of 512x512 pixels. The network is composed of five encoding and five decoding blocks, each block consisting of two 3x3 convolutions. A residual connection around the two convolutions was added to improve the learning capacity of the network (He et al., 2016). A batch normalization layer between each convolution accelerates the training and makes the training performance less dependent on the initial weights (Ioffe and Szegedy, 2015). On the encoding side, the blocks end with a 2x2 max pooling operation while on the decoding side, blocks start with a 2D upsampling (or "up-convolution") operation. The skip connections between the encoding and decoding blocks are implemented by taking the final convoluted map of each encoding block and concatenating it to the first map of the corresponding decoding block. The number of features in the first layer is 16, and the number of features is doubled with increasing layer depth. Additionally, dropout regularization is added between the two deepest convolution layers of the network in order to avoid over-fitting.

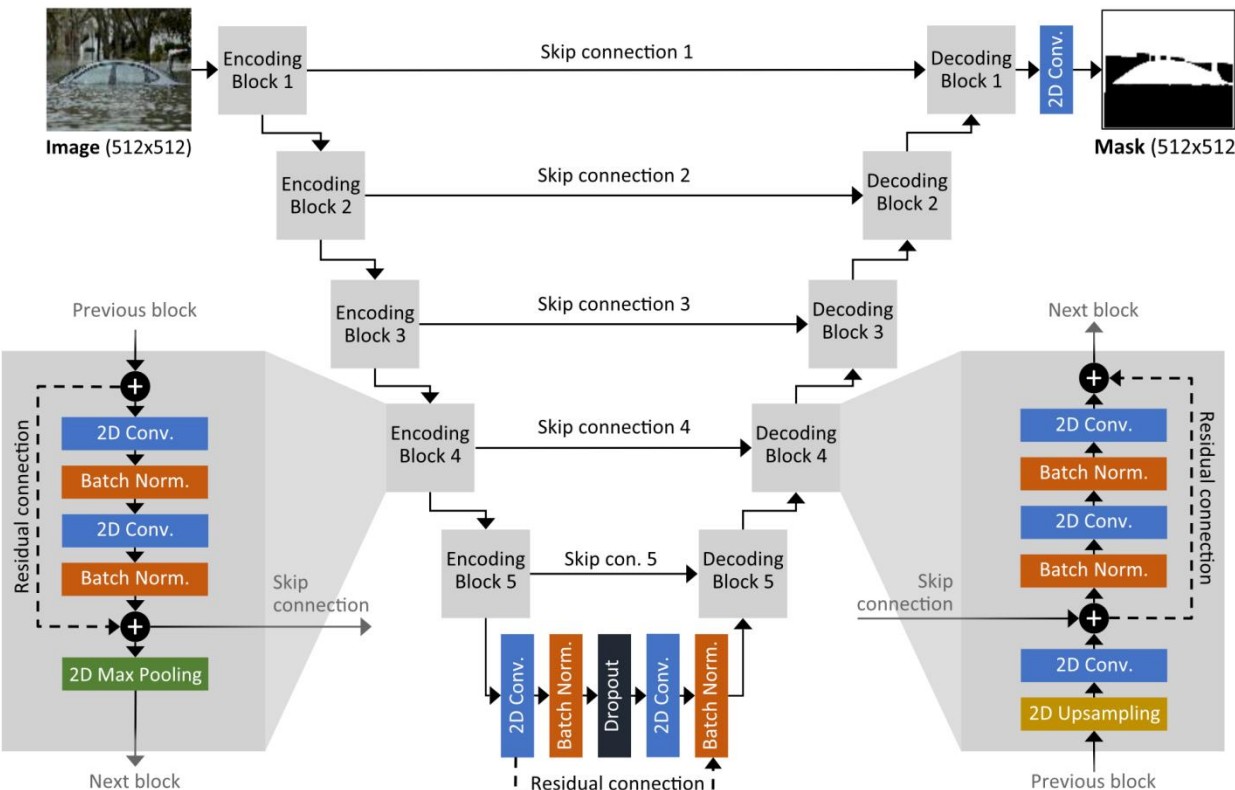

**Figure 2: The deep convolutional neural network architecture used to perform water segmentation.**

### 2.1.3 Deep convolutional neural network training strategies

The collection and labelling of training data is one of the most costly and time-consuming aspects of training DCNNs. For the specific application of flood detection in CCTV and webcam images, training images are particularly rare. Therefore, in this study we evaluated the effectiveness of two strategies for increasing segmentation performance with few training images.

Given the relative rarity of flooding images from surveillance cameras, we used a collection of 1218 labelled images that were collected from the Internet and manually labeled (Chaudhary, 2018). Since almost all of the images in the dataset are subject to copyright, we provide a sample of images in the public domain that are representative of the dataset in Fig. 3. These images have two differences as compared to typical surveillance camera images. First, the image quality is generally better in terms of resolution and color reproduction. Second, the pictures almost only depict extensive flooding where most of the ground is covered by water. To provide examples of dry ground, 300 images of street scenery without flooding from the Cityscapes dataset (Cordts et al., 2015) were added to the 1218 Internet images, forming a pool of reference images.

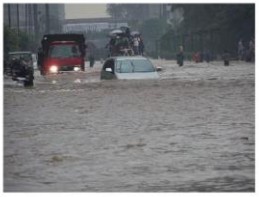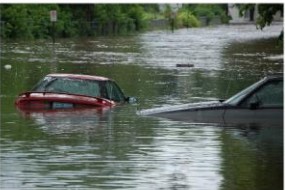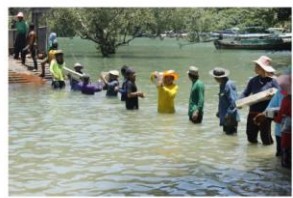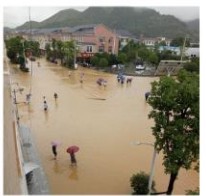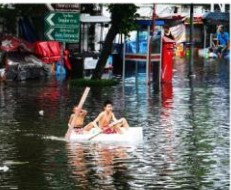

**Figure 3: Public-domain images of flooding, representative of the dataset provided by Chaudhary** (Chaudhary, 2018) **that was used in this work.**

These reference images were used to train a **Basic** version of the DCNN (80% for training, 10% for validation, and 10% for testing). Conventional augmentation was applied to the images as they were fed into the network: a random displacement of up to 20% and a random horizontal flip. In this work, we considered two strategies for improving the flood segmentation performance of the "Basic" training strategy (Tab. 1).

In the **Augmented** strategy, the same images as for the basic training strategy were used but with additional augmentation steps that degraded image quality to the level of typical surveillance footage. The following image transformations, implemented with the Keras library (Chollet and others, 2015) were applied during augmentation:

- random horizontal mirroring
- translate image horizontally and vertically by +/- 20%
- change in contrast by +/- 50%
- resolution deterioration by zooming into image at different locations up to 33%
- decreasing saturation by up to 80%
- alter image brightness between -80 and +20 units
- blurring with random Gaussian filter

Augmentation was applied with a 20% probability each time an image was fed into the network for training (up to 200 times, corresponding to the number of epochs).

In the **Fine-tuned** strategy, we performed transfer learning to adapt the DCNN to specific surveillance videos. This was done by retraining the "Augmented" network with seven manually labeled frames from each video.[1] The retraining is performed in two steps. First, only the weights of the last deconvolution block are released and retrained. Then, the rest of the weights are also released and the whole network is retrained. This process resulted in a distinct and specialized network for each footage sequence. In particular, it allowed the networks to learn specific camera characteristics that may not have been represented in the reference images. However, the additional effort required by this training approach limits its practical utility to situations where the increase in data quality is of particularly high value.

---

[1] Manual labelling takes around 2 minutes per image for someone with a little experience.

To train the network, the Adaptive Moment Estimation (Adam) was chosen as the gradient descent optimizer because it shows good convergence properties (Kingma and Ba, 2015). The dice coefficient served as the loss function, defined after Zou et al. (2004). The DCNN was trained on an Nvidia Titan X (Pascal) 12 GB GPU. The "Basic" strategy took approximately 120 minutes on average, whereas the "Augmented" strategy required approximately 180 minutes for training. The fine-tuning process required an additional 5 minutes of training per video.

## 2.2 Static Observer Flooding Index

The Static Observer Flooding Index (SOFI) is introduced in this work as a dimensionless proxy for water level fluctuations that can be extracted from segmented images of stationary surveillance cameras. The SOFI signal is computed as

$$\text{SOFI} = \frac{\#\text{Pixels}_{\text{Flooded}}}{\#\text{Pixels}_{\text{Total}}} \tag{1}$$

and corresponds to the visible area of the flooding as seen by a stationary observer. Its value can vary between 0% (no flooding visible) and 100% (only flooding visible). When this index is evaluated at multiple consecutive moments in time, the variation of its value provides information about fluctuations of the actual water level under the assumption that the camera remains static and that the view of the flooding is not overly obstructed by moving objects or people. In principle, objects and people will move in and out of the image at a higher frequency than the water level fluctuations, so the influence of such obstructions should be limited to an additional noise that can be filtered out. Nevertheless, situations may arise in which the assumption does not hold, for example if an obstruction is permanently removed from the scene during a flood event.

In certain cases, it may make sense to restrict the computation of the SOFI to a region of interest (ROI) of the image. For example, if the image contains more than one hydraulic process, such as accumulation in one part of the image and flow in the other, a ROI can be defined so that the SOFI only reflects the evolution of the accumulation process. The ROI can also be defined to exclude areas in which water segmentation is problematic due to unfavorable lighting conditions or visual obstructions, for example. Finally, the ROI can also be chosen over a region of the image in which changes of water level are going to be most visible, e.g. over a vertical wall. In this study, the ROI was implemented by means of a rectangular selection made by the authors according to these criteria. To gauge the effectiveness of this measure, performance was assessed both with and without user-defined ROI.

## 2.3 Performance assessment

### 2.3.1 Surveillance footage

Six videos depicting flooding were used to assess the performance of the proposed flood monitoring approach. Table 2 provides the characteristics of these videos, which provide a diverse and realistic range of environmental conditions and image qualities.

To assess the performance of the method in terms of flood trend extraction, a reference for the water level trend was established for each video. In the cases of the two FloodX camera videos, a water level signal was available from ultrasonic rangefinders, as documented in (Moy de Vitry et al., 2017). For the other four videos, a qualitative trend was visually estimated with an arbitrary scale. The qualitative trend obtained in this manner was judged sufficient for the present study since this study only investigates the ability of SOFI to predict water level trend, and not the actual water level.

### 2.3.2 Flood segmentation performance

Three images from each video, representing low, medium, and high flooding conditions, were used to assess segmentation performance. Image segmentation is a common classification task that is often evaluated with the mean intersection over union ratio (IoU), also known as the Jaccard index (Levandowsky and Winter, 1971). This metric is applied by running the DCNN on a series of testing images that were not seen during training, and comparing the segmentation result (S) to a manually annotated ground truth (G). The mean IoU is then computed as

$$\frac{1}{N}\sum_i^N \frac{|S_i \cap G_i|}{|S_i \cup G_i|} \tag{2}$$

where N is the number of testing images and $S_i$ or $G_i$ is the water-covered area in a segmented image or corresponding ground truth image, respectively. The index varies between 0% for complete misclassification and 100% for perfect classification.

### 2.3.3 Performance of SOFI as a proxy for the water level trend

To evaluate whether SOFI can be considered a proxy for real water level trends, one can assess to what extent the relationship between the two signals is monotonic increasing. This quality can be evaluated with the Spearman rank-order correlation coefficient (Spearman, 1904), which is used to measure the degree of association between two synchronous signals. Importantly, it does not assume any other (e.g. linear) relationship between the two signals. To compute the coefficient, the rank of each signal value must be computed relative to its respective signal. For signals in which the same value can appear multiple times (tied ranks), the Spearman rank-order correlation coefficient $\rho$ is given by

$$\rho = \frac{\sum(x_i-\bar{x})(y_i-\bar{y})}{\sqrt{\sum(x_i-\bar{x})^2\sum(y_i-\bar{y})^2}}. \tag{3}$$

where $x_i$ and $y_i$ are the ranks of the two signals for time step $i$, and where $\bar{x}$ and $\bar{y}$ are the average ranks of the SOFI and water level signals, respectively. In the current study, the reference signal for the water level trend was obtained either from an in-situ sensor or by visual inspection of the surveillance footage, as described in Section 2.3.1. The pandas Python library (McKinney, 2010), which contains an implementation of the Spearman rank correlation coefficient, was used for time series analysis.

# 3 Results

## 3.1 Automatic flood water segmentation

Image segmentation, with the setup described in Section 2.1.3, takes around 50ms per image. Figure 4 provides sample frames from three of the six surveillance videos used in this work. The other three videos are provided in Figure A1 of the supplementary material. In each case, the human labels as well as automatic segmentations from the various DCNNs are drawn in blue. Additionally, the ROIs are drawn in red for each video, defined manually according to the criteria mentioned in section 2.2.

The basic DCNN was able to detect water in certain cases, but also committed large segmentation errors in the cases of the *FloodXCam1*, *Garage*, and *Park* videos. In the *Parking lot* video, segmentation appears quite successful, which could be due to the scene being visually similar to the images with which the "Basic" network was trained.

Compared to the "Basic" DCNN, the "Augmented" DCNN provides a visible improvement in most cases. The case of *FloodXCam1* is an exception since the DCNN successfully segmented the shallow water flowing on the ground (which was not classified as flooding in the human labels), but did not detect the water ponding in the upper right of the image. This error is fixed in the "Fine-tuned" DCNN for this video thanks to the use of additional training images from the *FloodXCam1* video.

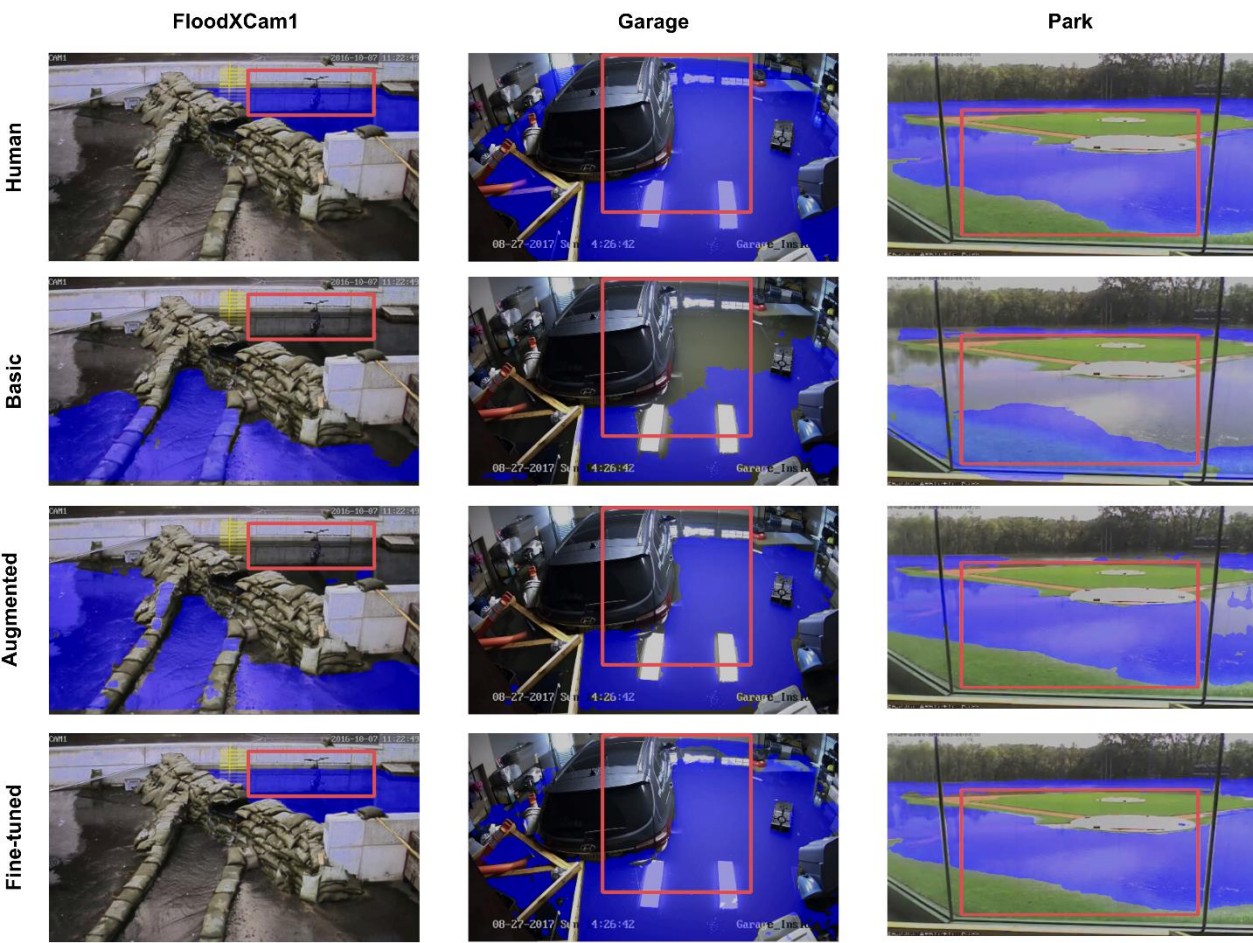

Figure 4: Sample frames taken from three of the six analyzed surveillance videos, shown with the human label or automatic flood segmentation in blue and the regions of interest (ROI) in red. The samples show how the "Augmented" strategy improves segmentation in the *Garage* and *Park* samples, but the segmentation for *FloodXCam1* is only successful with the "Fine-tuned" training strategy. The sample frames from the remaining three videos are provided in the supplementary materials.

Figure 5 shows the segmentation performance of the DCNNs measured by the IoU, both for the full image and within the defined ROI. The "Augmented" DCNN improves performance for all videos except for the two *FloodX* videos. In the case of the *Park* video, the improvement of IoU for the full image is around 30 percentage points. For the two *FloodX* videos, however, segmentation seems to suffer slightly under the "Augmented" network, possibly because the augmentation transformations increased dissimilarity of the training images instead of vice versa. For these two videos, improvement is only achieved thanks to fine-tuning, which proved beneficial for all videos by providing IoUs higher than 90% on average. Figure 5 also shows that within the "expert-defined" ROIs, segmentation performance is generally worse. We conclude that a human is generally not able to identify and exclude "difficult" regions of the image a priori, which was one of the original reasons for defining an ROI.

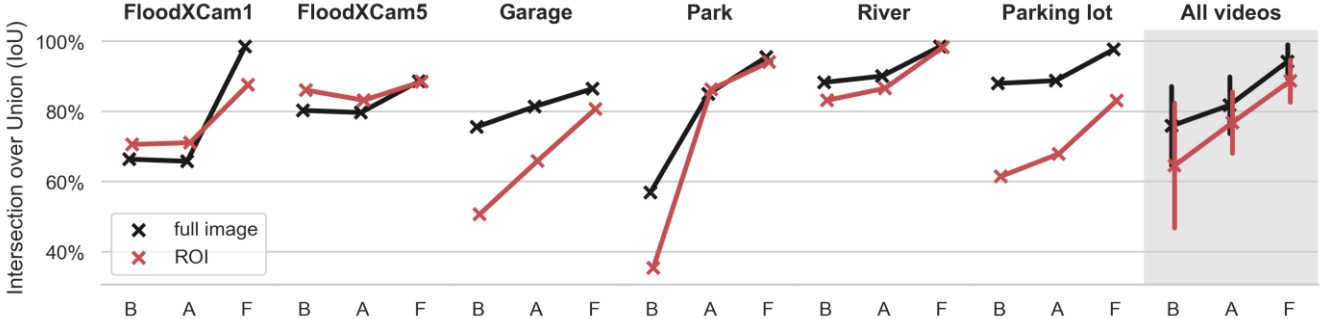

**Figure 5: Automatic segmentation performance measured in terms of the Intersection over Union (IoU) for each video using the "Basic" (B), "Augmented" (A), and "Fine-tuned" (F) DCNNs. On the right side, the overall performance for all videos is depicted, with the vertical segments representing the standard deviation of values.**

### 3.2 Flood level trend extraction

After the frames of a video are segmented, the SOFI is computed for each frame, providing information about the temporal evolution of the visible flood extent. In Fig. 6, the case of video *FloodXCam5* is a clear example of how the SOFI reflects changes in the actual water level. Comparing the SOFI signals to the measured water level, it is evident that a correlation exists, although the relationship does not appear linear. For all DCNN training strategies, the trend of the SOFI signal from the ROI (in red) is easier to visually identify than the trend of the SOFI computed from the whole image (in black), which is advantageous if the signal is to be visually interpreted. However, the SOFI signal from the ROI is also noisier, and does not capture the first flood event occurring at 12:48.

In Fig. 6, we also see that a systematic segmentation error is committed in that water is always falsely detected on the ground between the two events. This example illustrates why our approach focuses on the trend of the SOFI signal and not its absolute values, which are more sensitive to systematic errors.

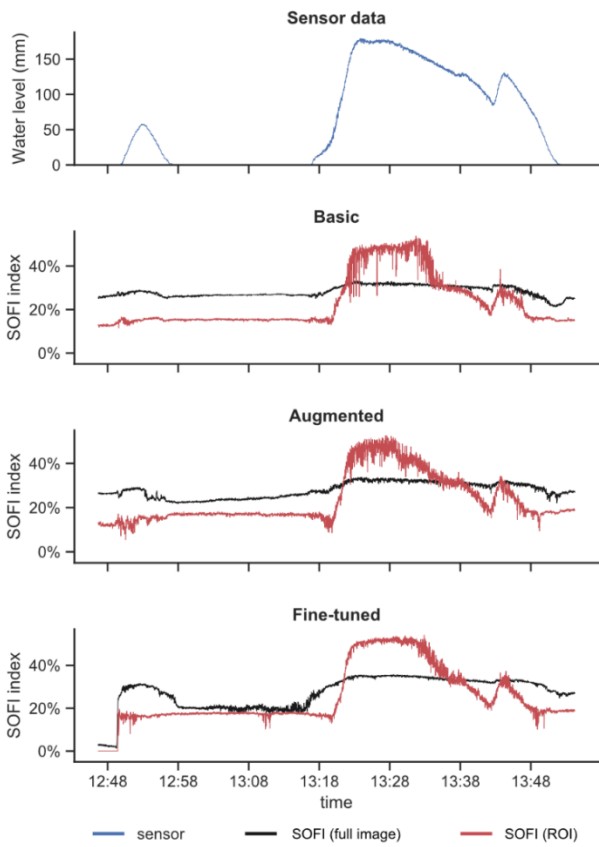

**Figure 6: Water level (blue) and SOFI signal for whole image (black) or Region of Interest (ROI, red) for video *FloodXCam5*. With more advanced DCNN training strategies, noise and outliers in SOFI are visibly reduced in the ROI. The time series for the remaining videos can be found in the supplementary materials, Figures A2-A6.**

In Fig. 7, the relationship between the SOFI and the water level is further explored for both *FloodX* videos. This figure shows how the relationship between SOFI and the water level can be non-linear, a consequence of the topography in which the flooding occurred. For example, in *FloodXCam5* a large area of the image is rapidly segmented as water starts covering the floor of the basement, causing an almost vertical segment on the left side of the scatter plot (especially visible for the

5 "Fine-tuned" network). For the SOFI computed from the full images, we also see systematic and time-variant errors, resulting in portions of the data having a larger internal correlation that are visible as strands of points in Fig. 7.

Generally, it appears that the use of an ROI (red) leads to a stronger association between water level and SOFI. However, the SOFI signals from the ROIs also contain more noise than the SOFI signal derived from the full image. Additionally, in *FloodXCam5*, it seems that the ROI was poorly selected, resulting in lower sensitivity of SOFI to the water level up to a

10 depth of around 100 mm.

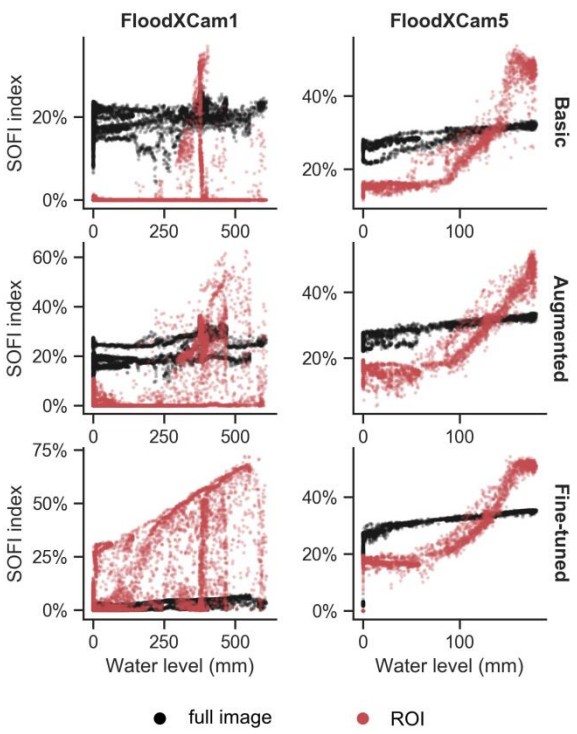

**Figure 7: Videos with reference measurements: SOFI computed from full images (black) or from ROI (red) plotted against measured water level for different training strategies. The reduction of noise and systematic errors with the "Fine-tuned" strategy is evident.**

Figure 8 shows the relationship between SOFI and the visually estimated flood intensity in the remaining videos, for which no in-situ water level measurement is available. In this figure, the value of the "Augmented" and "Fine-tuned" networks appears in the progressive reduction of noise in the SOFI signal.

5 Two exceptional cases in Fig. 8 need to be explained in more detail. First, the SOFI signal for the *River* footage suddenly appears to be arbitrary at the highest flood intensity. This is due to intermittent submersion of the camera by the floodwater that leads to gross segmentation errors. The *Garage* video is also exceptional; flooding caused objects to float around in the garage causing constant changes of the visible inundated area and thus noise in the SOFI signal.

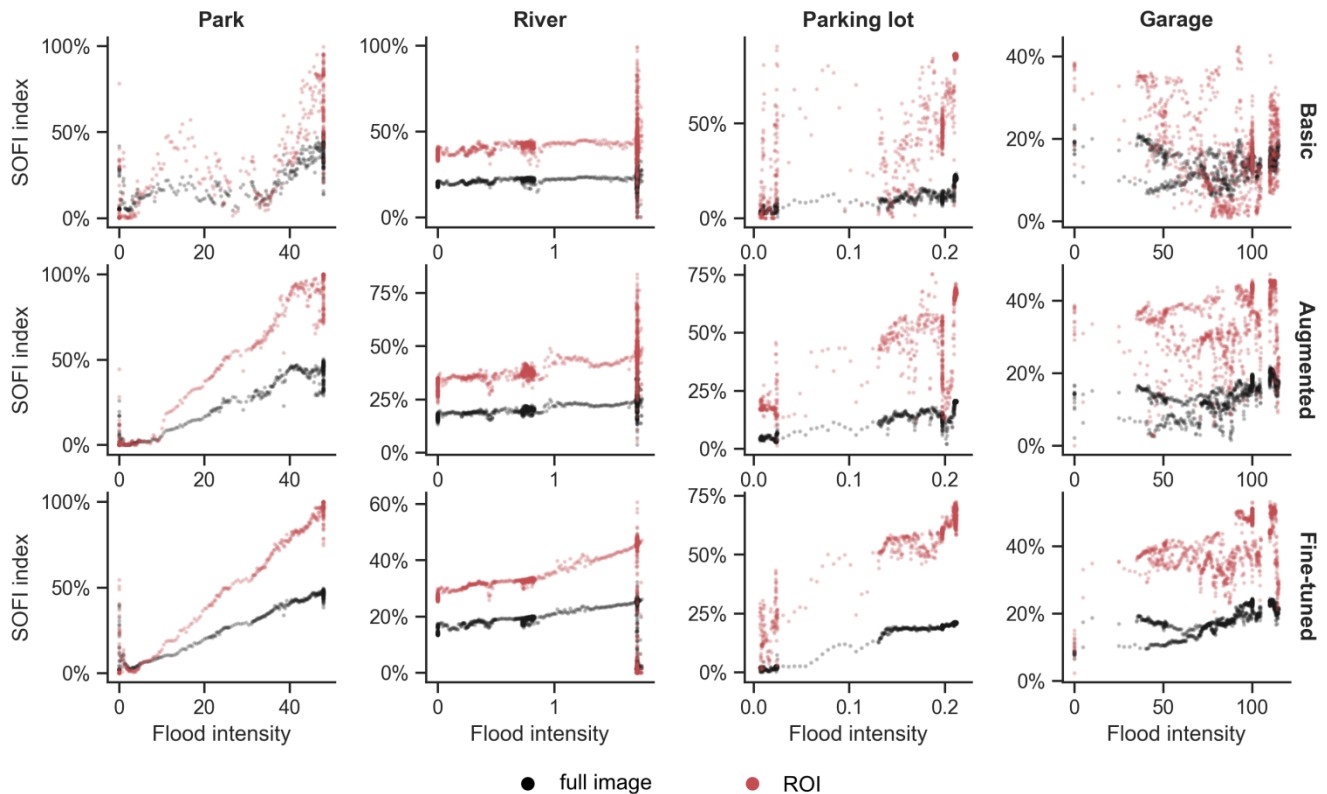

10 **Figure 8: Videos without reference measurements: SOFI computed from full images (black) or from ROI (red) plotted against visually estimated flood intensity, for different training strategies. There is a visible reduction of noise and increased correlation with the "Augmented" and "Fine-tuned" strategies.**

Figure 9 shows the Spearman correlation coefficients between the SOFI and the flooding intensity for each video and each training strategy. This figure shows that using the "Augmented" training strategy, the Spearman correlation coefficient for the full image reaches 75% on average, while for the "Fine-tuned" training strategy the average correlation coefficient reaches around 85%.

We draw two general conclusions from the results shown in Fig. 9. First, defining an ROI does not consistently improve the ability of the SOFI signal to reproduce flood trends. This could be due to the poorer segmentation performance within the ROIs (Fig. 5), thereby introducing noise in the water level-SOFI relationship (Fig. 7 and Fig. 8). Second, the "Fine-tuned" networks generally improve the correlation of SOFI with the water level trend, with the exception of the *River* video. As made visible in Fig. 8, the submersion of the camera leads to frequent outliers that become more distinct from the rest of the signal after fine-tuning, leading to a lower correlation with the water level trend.

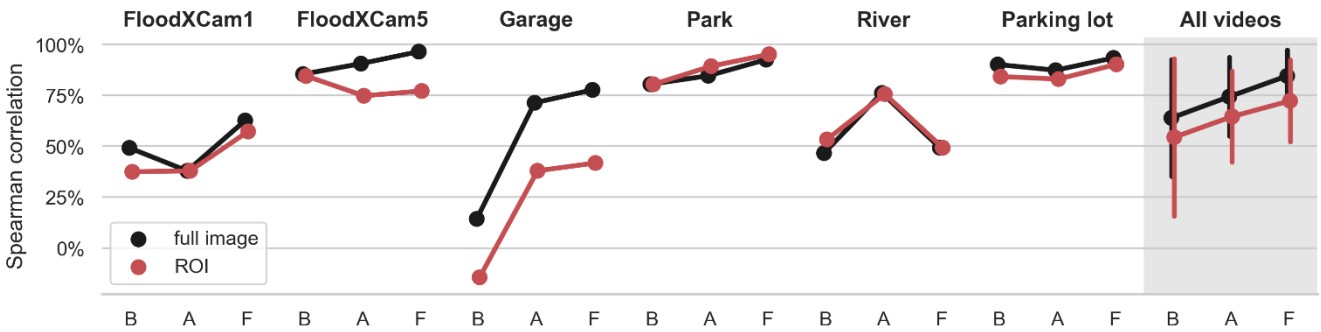

**Figure 9: Spearman rank correlation coefficients for each video using the "Basic" (B), "Augmented" (A), and "Fine-tuned" (F) DCNNs. The final category *All videos* excludes the *River* video which is an outlier because the camera is intermittently submerged. The vertical segments in the *All videos* category represent the standard deviation of values. The results show that both the "Augmented" and "Fine-tuned" networks improve the correlation of SOFI with the water level, although performance is still varied.**

## 4 Discussion

### 4.1 SOFI as a scalable and robust approach for qualitative water level sensing

Compared to alternative methods for water level monitoring, our approach is less ambitious in the type of information it aims to provide, as it only attempts to communicate water level fluctuation and not an absolute water level. This weakness is at least partially compensated for by an increased scope of applicability, as it is almost the only surveillance image-based monitoring approach designed to provide water level information without needing to be calibrated to each camera. As described in Section 1.3, research has typically been focused on extracting absolute water levels from images, mainly by identifying the water edge in relation to an object that has been measured and manually identified in the image (Bhola et al., 2018; Kim et al., 2011; Sakaino, 2016). Although Lo et al. (2015) present a method that does not necessarily need reference

measurements, human intervention is still needed for each camera to define virtual markers and a "seed" to inform water segmentation. The concept presented by Jiang et al. (2019) is an exception, theoretically overcoming the need for per-image calibration by taking ubiquitous objects as scale references. Nonetheless, that method assumes that such reference objects are visible in the flooding scene, which is an assumption that may not always hold. In comparison, SOFI does not need specific objects to be present, nor does it need field measurements or manual set-up of the image. The DCNN-based water segmentation approach used is flexible enough that given sufficient training data, it can theoretically be applied to any situation. In this study, the scalability of SOFI was demonstrated by applying the "Augmented" DCNN to six videos not seen during training, from which information was extracted despite complex environments, moving objects, and bad lighting conditions.

The simplicity of the SOFI approach also gives it robustness. By using areal integration to quantify flood intensity, the index is less sensitive to small segmentation errors, which could be problematic if punctual virtual markers are used as in the case of Lo et al. (2015). Also, since SOFI only claims to provide trend information, systematic segmentation errors (e.g. misclassification at the water-background boundary) should only be of minor consequence. This aspect of SOFI is well illustrated in this study by the *Park* video, for which the SOFI signal had a high correlation to the water level trend despite mediocre segmentation performance.

Besides the advantages of SOFI, its use of a DCNN carries a disadvantage due to its need for training data. Methods that use heuristics to segment water, such as Lo et al. (2015), do not require as much training data, although their application is then often limited to visually similar situations.

## 4.2 Factors influencing the information quality of SOFI

Although SOFI is theoretically applicable at locations where the extent of visible water correlates with flooding depth, there are certain conditions that can negatively influence the quality of information obtained.

First, floodwater can look very different from case to case. For example, (i) water surface movement can cause a range of different wave structures, (ii) the color of the water is variable and in very agitated flows, bubbles can make the water appear white, (iii) water reflects light that falls on it, at times in a mirror-like fashion, and (iv) surveillance cameras tend to have low color fidelity, dynamic range, resolution, and sharpness. Due to the complexity of segmenting floodwater, one can expect that more training images would be required than for a typical segmentation problem. In our results, the high inter-video variability in segmentation performance suggests that the number of training images should be increased in future studies. In particular, the variability suggests that the training images were not fully representative of the testing images. Thus, it is probable that the segmentation performance could be substantially improved if a larger and more diverse training set were available.

Second, in situations where large quantities of manually labeled images are required, it is inevitable that some labelling errors occur. Looking into the implications of such errors, Heller et al. (2018) showed that U-net is relatively insensitive to

jagged label boundaries. Regarding non boundary-localized errors, all DCNN architectures investigated were found to be very robust.

Finally, the basic assumption that the extent of visible water correlates with flooding depth may not fully hold for every camera scene. For example, small floods may not be visible due to obstructions, and if the camera is oriented such that the whole image can be covered by water, the highest stages of flooding may be censored from the signal as well. These situations will require special handling when the data is assimilated for model calibration. The scene topography and camera placement also affect the slope and linearity of the SOFI-water depth relationship, and can make the trend more difficult to determine if the correlation in the relationship is small. Additionally, the entry and exit of objects in a flooding scene can compromise the approach, especially if the objects are large and occur with low frequency. The flooding itself may increase the occurrence of such obstructions, for example in the case of cars stuck in traffic or objects transported by the water.

## 4.3 Degradation of training image quality improves segmentation of surveillance images

In this study, the training images available were of higher quality than typical images originating from surveillance cameras, a discrepancy that was expected to limit the performance of the DCNN on surveillance images. Therefore, in the "Augmented" training strategy, the augmentation step included transformations that lowered the quality of the training images, making them more similar to surveillance images.

The results from the six videos used in this study confirm that the artificial degradation of training image quality not only improves segmentation performance in surveillance footage but also increases the correlation of SOFI with the actual water level trend. While in most videos the improvement was clear, no improvement was observed for the two *FloodX* videos. These two videos stand out in terms of low image quality, location of water in the upper part of the image, and a different setting surrounded by concrete walls. We therefore see a need to investigate such failure cases in order to improve the training data collection and augmentation steps.

## 4.4 Fine-tuning of the DCNN to specific cameras

In situations where segmentation performance is critical, one can fine-tune a general DCNN to a specific surveillance camera. Even with very few additional training images, we find that the segmentation performance and correlation of SOFI with water level trends both improve thanks to fine-tuning. Despite this result, it should be kept in mind that the fine-tuned DCNNs also lose some generality and may, for example, have issues when lighting conditions are different from those present in the images for fine-tuning.

Our recommendation for fine-tuning is that care should be taken in creating a set of training images that is roughly as diverse as situations in which the fine-tuned DCNN will be used. Additionally, although we performed fine-tuning with very small sets of seven images per video, fine-tuning performance could be further increased by using more images.

### 4.5 Regions of interest (ROI) do not deliver expected value

The definition of ROIs for SOFI computation, motivated by the possibility of omitting difficult portions of the images and focusing on more information-rich portions, proved unsuccessful. Not only was it difficult for the human "expert" to foresee what area of the image met the above criteria, but a systematic increase of noise in the SOFI signal was observed within the ROI. We therefore do not recommend that ROIs be defined systematically for all cameras, but only in cases where multiple hydraulic processes are visible in the image and need to be distinguished.

### 4.6 Practical value of SOFI

Since the qualitative nature and noisiness of SOFI might raise questions about its practical value, two aspects must be recalled. First, SOFI aims to provide information in the context of urban pluvial flood events, for which monitoring data is admittedly difficult to obtain. The studies cited in the introduction have proven that in situations of data scarcity, even qualitative information can be useful in improving model accuracy. In particular, the sparse stream level class information used by van Meerveld et al. (2017) is conceptually equivalent to the flood level trend information contained in SOFI. Second, thanks to the scalability of SOFI, one should be able to apply it to large surveillance networks or retroactively to archived footage at a reasonable cost. By providing data for model calibration and validation, the approach we propose can help reduce modeling uncertainty for urban flood risk assessment and assist in the planning of flood mitigation measures. In addition, the flood level trend information provided by SOFI could help direct the focus of decision-makers and rescue personnel during flood events so that resource use can be optimized. For insurance companies, the flooding information can also help verify claims and establish fair insurance policy premiums.

### 4.7 Recommendations for future research

Future research should assess the actual value of the information provided by SOFI for the validation and calibration of urban flood models. For this, it would be of value to have a large-scale, long-term case study of a flood-prone urban area, in which surveillance footage and reference flood measurements are available. Additionally, possible methods to de-noise the SOFI signal and quantify its reliability should be investigated. In particular, the issue of rapid changes in the SOFI signal due to moving obstructions could be addressed by filtering changes that surpass a given threshold. Another approach to this problem is through automatic definition of ROIs that censor out noise-generating areas of the video scene.

### 5 Conclusions

In this study, we explored the potential of using a deep convolutional neural network (DCNN) and a simple but novel index (SOFI) to obtain flood level trend information from generic surveillance cameras. The results of our study strongly suggest that qualitative flood level information can indeed be extracted automatically and universally from any static camera, although we see the need for many training images to cover the range of appearances that floodwater can take. To

compensate for the limited number of training images available in this study, we found that degrading image quality during training improved segmentation performance by approximately 10% (IoU) on low-quality surveillance images. Additionally, fine-tuning the DCNN to a specific video with as few as seven manually labeled images further improved performance. In our results, the SOFI signal from the camera-independent DCNN correlated with water level trends at a rate of 75% on average (Spearman rank correlation coefficient). The proof of concept presented in this study has significant implications. Namely, it confirms that footage from surveillance cameras, processed with artificial intelligence, could provide urban flood monitoring data for entire cities while preserving privacy. Without the need for specialized instrumentation or field measurements, monitoring could be conducted at a relatively low cost, which is especially attractive for cities with limited financial resources. Previous research suggests that the qualitative information contained in SOFI can easily be used to reduce parameter uncertainty in urban flood models. Apart from modelling-related applications, SOFI could also be useful for coordinating flood response and verifying insurance claims.

**Code availability**

The code used in this work for creating, training, and evaluating the DCNN, as well as extracting the SOFI and plotting results can be found in the following archive: https://doi.org/10.25678/000150.

**Data availability**

The licenses to the images and video data used in this work are held by third parties and cannot be republished by the authors. We encourage the interested reader to refer to the references provided for the individual data sources. The DCNN weights trained for each training strategy are available in the following archive: https://doi.org/10.25678/000150.

**Author contribution**

M.M.d.V. conceived the method; S.K. and M.M.d.V. implemented the method, analyzed the data, and made the figures; J.W. contributed guidance for the DCNN methods and J.P.L. was the principle investigator of the study, M.M.d.V. wrote the paper. All authors read and helped edit the manuscript.

**Competing interests**

The authors declare no conflict of interest. The funding sponsors had no role in the design of the study; in the collection, analyses, or interpretation of data; in the writing of the manuscript, and in the decision to publish the results.

## Acknowledgements

The authors thank Priyanka Chaudhary for providing the images and annotations used to train the DCNN, as well as Nadine Rüegg for DCNN implementation tips. This project was financed by the Swiss National Science Foundation under grant #169630.

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

**Table 1: Training strategies for the deep convolutional neural network**

| Strategy | Images used for training | Data augmentation steps | Training steps |
|---|---|---|---|
| **Basic** | 1214 images from Internet (Chaudhary, 2018) + 300 images from Cityscapes (Cordts et al., 2015) | Random displacement of up to 20%, random horizontal flip | End-to-end training with 80% of images for up to 200 epochs |
| **Augmented** | Same as "Basic" | Same as "Basic" + random Gaussian blur, color desaturation, contrast modification, brightness alteration, reduction of resolution | Same as "Basic" |
| **Fine-tuned** | Same as "Basic" + seven frames taken from each video | No augmentation is applied when fine-tuning the network. | Retrain "Augmented" network with 7 manually labeled video frames |

**Table 2: surveillance footage used in study**

| Video | Setting | Flood type | Image resolution | Water level trend | Analysis frequency (frames/min) | Event duration | Visual clarity of flooding | Source |
|---|---|---|---|---|---|---|---|---|
| **FloodX Cam1** | Experimental facility, weir | Artificial | 1280x720 | Sensor | 60 | 164 minutes | High: Wet surfaces | (Moy de Vitry et al., 2017) |
| **FloodX Cam5** | Experimental facility, cellar | Artificial | 1280x720 | Sensor | 60 | 166 minutes | Medium: Wet surfaces, desaturated image | (Moy de Vitry et al., 2017) |
| **Garage** | Indoor, garage | Extreme rainfall | 1280x720 | Visual | 24 | 19.8 hours | High: Moving objects floating in water | (Roisman, 2017) |
| **Park** | Outside, sport field | River flooding | 640x480 | Visual | 1 | 13.5 hours | Low: Camera behind window with reflections; camera is sometimes jostled; many | (Cityofchaska, 2010) |

| | | | | | | | | |
|---|---|---|---|---|---|---|---|---|
| | | | | | | | different lighting conditions throughout the day. | |
| **River** | Outside, river under concrete bridge | Extreme rainfall | 1280x720 | Visual | 30 | 6 hours | Low: Camera sometimes submerged; lens blurry due to water drops; high image compression | (Hurricanetrack, 2015) |
| **Parking lot** | Street Scenery | Extreme rainfall | 1280x720 | Visual | 12 | 200 minutes | Low: partially nighttime images; lens blurry due to water drops | (Blanchard, 2017) |