# Peer review of "Scalable Flood Level Trend Monitoring with Surveillance Cameras using a Deep Convolutional Neural Network"

_Hydrology and Earth System Sciences, 2018_

## Referee Comment (RC1) · Anonymous Referee #1 · 14 Mar 2019

Overall, this is a thoroughly interesting and relevant piece of work. The lack of existing techniques to quantify urban flood distribution in a quantitative manner provides significant challenges for urban drainage engineering, and as the authors themselves note, acquisition of data is nigh-on-impossible using conventional techniques.

This paper presents a unique alternative, the capture and utilisation fo CCTV imagery, to track the distribution and depth of flood events and via the implementation of a Deep Convolutional Neural Network, provides scope to automate the process through a machine learning procedure. This would be an exceptional contribution to the literature in this area and I would definitely like to see this published.

[Figure]

Several areas perhaps need some addressing before the paper were to be published, but I hope these are relatively minor comments! Given the target audience will primarily be hydrologists, I find the description of the methodology (particularly the software development and computational methodology) to be very complex and in-depth. I'm sure for some in the community this will be fairly easy language to interact with, but certainly to someone that doesn't design, code or work the tools such as this, I found much of the section passed straight over my head.

The presentation of some of the figures are a little poor - while the matrix of images used to highlight how the model has been trained are undoubtedly a worthwhile addition to the paper, I definitely found them a little illegible due to their size - even zooming right in to the PDF rendered some of the images unclear as to what was going on. Furthermore, I originally elected to print a copy of the paper to review while travelling and the images do not translate to black and white or greyscale, perhaps highlighting an accessibility issue.

Appreciating the author's comments about copyright and third party ownership of CCTV imagery, I do feel a specific case study may have aided the paper rather than just the catalogue of training videos presented in section 3.1. I understand this may not be an option for the authors, based on the nature of the copyright imagery they had access to, but it would have been useful to see an example using real data and images from an observed urban flood event.

Finally, I found the discussion section to be a little too focused on the technical aspects and challenges of the methodology. What would have been nice, was a slightly more "big picture" discussion that highlight the impacts and possible extensions of this work in an urban setting. Some of the findings here would be significant for urban drainage managers, infrastructure managers (particularly transport etc), quantifying economic risks associated via damage etc and future policy creation around climate change scenarios and drainage dimensions. I acknowledge the authors touched on some of these issues in the introductory section, but it would have been a nice conclusion to the paper

to discuss how some of their results may help remove some of the challenges that they identified in the introduction.

---

## Author Comment (AC1) · 28 Mar 2019

**1  General reply to comments from Referee #1**

We thank Anonymous Referee #1 for the positive appraisal of the manuscript and valuable suggestions.

[Figure]

**2 Comment #1.1**

*Given the target audience will primarily be hydrologists, I find the description of the methodology (particularly the software development and computational methodology) to be very complex and in-depth. I'm sure for some in the community this will be fairly easy language to interact with, but certainly to someone that doesn't design, code or work the tools such as this, I found much of the section passed straight over my head.*

Thank you for this notice. We are aware that this manuscript and the description is outside the discipline of pure hydrology. After considering moving the methodological details to the appendix, we are inclined to leave them in the main body of the article. The reason we find this appropriate is that the methodology presented forms the core novelty of the paper.

**3 Comment #1.2**

*The presentation of some of the figures are a little poor - while the matrix of images used to highlight how the model has been trained are undoubtedly a worthwhile addition to the paper, I definitely found them a little illegible due to their size - even zooming right in to the PDF rendered some of the images unclear as to what was going on. Furthermore, I originally elected to print a copy of the paper to review while travelling and the images do not translate to black and white or greyscale, perhaps highlighting an accessibility issue.*

Thank you for raising this point.

**Changes**: We will change Figure 4 to improve its readability: we are planning to move the three less interesting case studies to the appendix, so that the size of the images can be increased. Also, we will increase the image resolution to 600 dpi. Finally, we

will modify the colors so that the segmentation boundaries are clear.

**4 Comment #1.3**

*Appreciating the author's comments about copyright and third party ownership of CCTV imagery, I do feel a specific case study may have aided the paper rather than just the catalogue of training videos presented in section 3.1. I understand this may not be an option for the authors, based on the nature of the copyright imagery they had access to, but it would have been useful to see an example using real data and images from an observed urban flood event.*

It is true that a specific case study of a real urban flood event would be beneficial to show the practical feasibility of the method proposed. Unfortunately, the possibility to collect such data would require a long-term study that was not within the means of the project. Nevertheless, other projects that aim to collect real flooding data could pick up this point. For example, the data collected in the ongoing Flood-Prepared project (link) might be a good candidate for such an applied study.

**Changes**: We will mention the need for a more in depth, practical case study of urban flooding in the discussion section of the manuscript.

**5 Comment #1.4**

*Finally, I found the discussion section to be a little too focused on the technical aspects and challenges of the methodology. What would have been nice, was a slightly more "big picture" discussion that highlight the impacts and possible extensions of this work in an urban setting. Some of the findings here would be significant for urban drainage managers, infrastructure managers (particularly transport etc), quantifying economic*

*risks associated via damage etc and future policy creation around climate change sce-*
*narios and drainage dimensions. I acknowledge the authors touched on some of these*
*issues in the introductory section, but it would have been a nice conclusion to the paper*
*to discuss how some of their results may help remove some of the challenges that they*
*identified in the introduction.*

Thank you for this suggestion.

**Changes**: In the conclusions section of the manuscript, we will elaborate on the ad-
vances that the proposed method could make possible. We will do this at the same
"big-picture" level as the introduction.

---

## Referee Comment (RC2) · Anonymous Referee #2 · 2 Apr 2019

Thank you for inviting me as a reviewer for the manuscript titled Scalable Flood Level Trend Monitoring with Surveillance Cameras using a Deep Convolutional Neural Network.

General comments: This paper proposed an approach to monitor flood level trend using DCNN. The topic is very interesting. However, in my opinion, it could be difficult for modelers, decision-makers and city planners to use the Static Observer Flooding Index (SOFI) directly. The authors should clearly explain what the direct or specific application scenarios of SOFI are. If SOFI or visible area of the flooding can be converted into water depth value or even class information on water depth, it would make

this approach more useful and valuable. Unfortunately, this paper lacks this attempt, so I suggest rejecting this paper but encourage resubmission after the improvements have been made.

Some specific comments are reported below: 1. Page 1, Line 16 "The results suggest that the approach can be used with almost any surveillance footage ". I think this conclusion may be too strong.

2. Some new related references about automatic water level monitoring with surveillance images should be included to strengthen this paper. Wang, R. Q., Mao, H., Wang, Y., Rae, C., & Shaw, W. (2018). Hyper-resolution monitoring of urban flooding with social media and crowdsourcing data. Computers & Geosciences, 111, 139-147. Jiang, J., Liu, J., Cheng, C., Huang, J., & Xue, A. (2019). Automatic Estimation of Urban Waterlogging Depths from Video Images Based on Ubiquitous Reference Objects. Remote Sensing, 11(5), 587. Bhola, P. K., Nair, B. B., Leandro, J., Rao, S. N., & Disse, M. (2019). Flood inundation forecasts using validation data generated with the assistance of computer vision. Journal of Hydroinformatics, 21(2), 240-256.

3. Page 4, Figure 1. There should be a "surveillance images" box between "camera" box and "deep conv. network" box.

4. Page 4, Line 18. Why U-net was selected for water segmentation?

5. Page 6, Lines 18-20. In the augmented strategy, how many images were used to train the U-net model? Is the augmented data the same amount as the basic data? Moreover, Page 7, Table 1, the augmentation steps in the augmented strategy should be clearly explained.

6. Page 7, Lines 5-6. It is not accurate enough. How many seconds or minutes does each model training take?

7. Page 8, Table2. The total frames or minutes and the resolutions of surveillance images should be given. How to define the quality of surveillance footage should be

explained.

8. Please add the average segmentation time of a single frame for the testing as a performance, since the real-time monitoring is usually important.

9. Page 9, Lines 16-17. The authors should explain more extensively how to compute the rank of each signal value.

10. Page 9, Line 20. "the two signals" should be clearly identified as the SOFI signal and the reference signal.

11. Page 12, Figure 6. Only the case of video FloodXCam5 was given, please add cases of parking lot and park that are the typical scenes of urban flooding.

12. Page 14, Line 2. "the visually estimated flooding intensity" and "Flooding intensity" in Figure 8 should be consistent with "the visually estimated water level" in the caption of Figure 8.

13. In the "Discussion" section, more comparisons with other existing methods for urban flooding information extraction from surveillance images should be added.

14. The accuracy of this approach could be affected by camera inclination and manual labeling. Topography can also affect the accuracy, e.g., in flat areas, the visible area of the flooding varies greatly, but the change of water depth value may be very small, while in low-lying areas, the change of visible area of the flooding is small, but the change of water depth value may be very big. These error sources should be discussed in the "Discussion" section.

15. In the "Conclusions" section, the authors should explain the direct applications and managerial implications of this approach.

---

## Author Comment (AC2) · 5 Apr 2019

**1  Reply to general comments from Referee # 2**

*This paper proposed an approach to monitor flood level trend using DCNN. The topic is very interesting. However, in my opinion, it could be difficult for modelers, decision-makers and city planners to use the Static Observer Flooding Index (SOFI) directly. The authors should clearly explain what the direct or specific application scenarios of SOFI are. If SOFI or visible area of the flooding can be converted into water depth value or even class information on water depth, it would make this approach more*

[Figure]

*useful and valuable. Unfortunately, this paper lacks this attempt, so I suggest rejecting this paper but encourage resubmission after the improvements have been made.*

We thank Referee 2 for the overall positive review, for the useful recommendations, and for the questions which will allow us to further improve the manuscript.

**On the value of flood level trend information**

While we agree with Referee 2 that decision-makers and city planners may not be accustomed to flood trend information, modelers in research have already successfully used trend data for model calibration (e.g. van Meerveld et al 2017), albeit in the field of natural catchment hydrology. In that publication, stream level class data are used as trend information in a conventional calibration scheme, with the minor adaptation being the use of the Spearman rank correlation coefficient as an objective function. Similarly, Wani et al. (2017) showed that even simple binary information of a combined sewer overflow in an urban catchment can reduce model parameter uncertainty. Making the link to our manuscript, binary flooding information can easily be derived from the flood trend information that SOFI provides, by defining for each camera a baseline SOFI value below which no flooding is assumed.

We agree with Referee 2 that our manuscript could better explain how the utility of SOFI is supported by literature. To do so, we will augment our manuscript with details and background information for the examples listed above. We will make these changes in the introduction, discussion, and conclusion of the manuscript.

**On the advantages of SOFI over water depth estimation**

Referee 2 also suggests converting SOFI into water depths (either values or classes)

if possible. Although methodologies for determining flood water levels are of value, we argue that the concept of flood trend monitoring with SOFI has advantageous characteristics that would be lost if SOFI were to be converted into a water depth.

1. **Higher scalability**. The obtention of water depth information from a segmented flooding image requires knowledge of the dimensions of reference objects in the camera scene. As shown by the literature suggested by Referee 2, there are different ways of obtaining this information. For example, one can rely on the visibility of ubiquitous objects of known dimensions (Jiang et al. 2019). This approach is limited to scenes in which such objects are visible, and fails if the reference object is hidden or partially hidden by mobile objects during a flood. Another way of obtaining reference measurements is with a survey of a large feature in each scene. Bohla et al. (2018) demonstrate this method nicely with bridges in urban stream settings. Needless to say, this method requires time, effort, and field knowledge for each of the sites surveyed, making it expensive and time-consuming to deploy at scale. SOFI was conceived as a metric that has few prerequisites and relies on minimal on-site knowledge, making it applicable to a broader range of scenes than any alternative method known to the authors.

2. **Fewer errors**. In stream or river settings, direct conversion of SOFI to a water depth is most probably possible e.g., by determining a "SOFI-water depth" curve analogous to an H-Q curve. However, in an urban environment, the movement of objects (e.g., cars, people) constantly change the extent of visible water. It follows that the "SOFI-water depth" curve will be inherently wrong. By setting lower ambitions, SOFI also relies on fewer assumptions about the scene monitored, and thereby introduces fewer possible sources of error.

3. **Lower risk of overvaluation**. Although a conversion of SOFI into water depth (either absolute value or class) could make the data more interpretable, it would

dissimulate uncertainty about the information delivered and could risk being valued at the same level of trust as data from conventional sensors. We therefore see value in keeping SOFI an adimensional, qualitative metric.

**Summary**

In summary, literature supports the argument that the flood level trend information reflected in SOFI is valuable in itself to improve urban flood analysis, and do not necessarily need to be converted into water depth. Furthermore, the conversion of SOFI into a water depth would not only reduce the ease of applicability, but also introduce additional error sources and lead to possible overvaluation. Based on these arguments, we do not see the need to transform SOFI into a water level estimation method. However, we understand that the value of SOFI can be clarified and we will carry out the changes described above to this regard. We will also present the advantageous characteristics of SOFI in an extended comparison with alternative methods, as requested by Referee 2 in Comment # 2.13.

**References**

Bhola, Punit Kumar, Bhavana B. Nair, Jorge Leandro, Sethuraman N. Rao, and Markus Disse. 2018. "Flood Inundation Forecasts Using Validation Data Generated with the Assistance of Computer Vision." *Journal of Hydroinformatics* 21 (2): 240–56. https://doi.org/10.2166/hydro.2018.044.

Jiang, Jingchao, Junzhi Liu, Changxiu Cheng, Jingzhou Huang, and Anke Xue. 2019. "Automatic Estimation of Urban Waterlogging Depths from Video Images Based on Ubiquitous Reference Objects." *Remote Sensing* 11 (5): 587. https://doi.org/10.3390/rs11050587.

van Meerveld, H. J. I., Vis, M. J. P., and Seibert, J.: Information content of stream level class data for hydrological model calibration, Hydrol. Earth Syst. Sci., 21, 4895-4905, https://doi.org/10.5194/hess-21-4895-2017 , 2017.

Wani, Omar, Andreas Scheidegger, Juan Pablo Carbajal, Jörg Rieckermann, and Frank Blumensaat. 2017. "Parameter Estimation of Hydrologic Models Using a Likelihood Function for Censored and Binary Observations." *Water Research* 121 (September): 290–301. https://doi.org/10.1016/j.watres.2017.05.038 .

**2 Replies to specific comments from Referee # 2**

**2.1 Comment # 2.1**

*Page 1, Line 16* "*The results suggest that the approach can be used with almost any surveillance footage* ". *I think this conclusion may be too strong.*

Thank you for pointing this out. This conclusion is indeed too strong given the limited number of videos on which the method was demonstrated (even though we chose videos that are diverse from a quality and scenery standpoint).

**Changes**: We will instead write that the results confirm that the method is versatile and can be applied to a variety of camera models and flooding situations.

**2.2 Comment # 2.2**

*Some new related references about automatic water level monitoring with surveillance images should be included to strengthen this paper. Wang, R. Q., Mao, H., Wang, Y., Rae, C., & Shaw, W. (2018). Hyper-resolution monitoring of urban flooding with social media and crowdsourcing data. Computers & Geosciences, 111, 139-147. Jiang, J., Liu, J., Cheng, C., Huang, J., & Xue, A. (2019). Automatic Estimation of Urban Water-*

*logging Depths from Video Images Based on Ubiquitous Reference Objects. Remote Sensing, 11(5), 587. Bhola, P. K., Nair, B. B., Leandro, J., Rao, S. N., & Disse, M. (2019). Flood inundation forecasts using validation data generated with the assistance of computer vision. Journal of Hydroinformatics, 21(2), 240-256.*

Thank you for referring us to these recent publications, which both highlight the need for flood monitoring data and illustrate the complexity of extracting absolute water level information from images.

**Changes**: We will add references to these publications to Sections 1.1 and 1.3 of the manuscript

2.3   Comment # 2.3

*Page 4, Figure 1. There should be a "surveillance images" box between "camera" box and "deep conv. network" box.*

Thank you for this input. We had several versions of Figure 1, some of which contained the box proposed, but left it out in the final version.

**Changes**: we will reintroduce a "surveillance video" box between "camera" box and "deep conv. neural network" box.

**2.4 Comment # 2.4**

*Page 4, Line 18. Why U-net was selected for water segmentation?*

U-Net is a very well known DCNN architecture for semantic segmentation, as the method has nearly 5000 citations in Google Scholar. It is well-suited to the flooding segmentation problem because of its relatively compact size compared to more recent state-of-the-art architectures (such as Mask-RCNN). The smaller size makes it both easier to train with small datasets (which we have) and faster to run, which is useful for flood monitoring.

**Changes**: We will include these reasons in the manuscript.

**2.5 Comment # 2.5**

*Page 6, Lines 18-20. In the augmented strategy, how many images were used to train the U-net model? Is the augmented data the same amount as the basic data? Moreover, Page 7, Table 1, the augmentation steps in the augmented strategy should be clearly explained.*

Thank you for this question. For the augmented strategy, we use the same images as for the basic strategy. Like for the basic strategy, each training image is fed into the network up to 200 times during training (fewer if the training is completed faster). For the augmented strategy, however, each image is first randomly transformed before being fed into the DCNN.

**Changes**: We will add information about the augmented strategy in Section 2.1.3. Furthermore, we will provide more details about the augmentation transformations applied.

**2.6 Comment # 2.6**

*Page 7, Lines 5-6. It is not accurate enough. How many seconds or minutes does each model training take?*

**Changes**: At the line indicated, instead of a range we will indicate the approximate average training time in minutes for the basic and augmented strategies separately.

**2.7 Comment # 2.7**

*Page 8, Table2. The total frames or minutes and the resolutions of surveillance*

*images should be given. How to define the quality of surveillance footage should be explained.*

It is true we could have provided more information here about the videos and the quality terms used.

**Changes**: We will add the requested information to Table2 and define quality in a table note.

**2.8 Comment # 2.8**

*Please add the average segmentation time of a single frame for the testing as a performance, since the real-time monitoring is usually important.*

Thanks for this request; it is true that this figure could be of interest to readers.

**Changes**: We will provide average segmentation time in Section 3.1

**2.9 Comment # 2.9**

*Page 9, Lines 16-17. The authors should explain more extensively how to compute the rank of each signal value.*

Thank you for your interest. The computation of the Spearman rank-order correlation coefficient, including signal rank of each value, was performed using the pandas Python library (McKinney, 2010), which is open source.

**Changes**: We will add a reference to this library in Section 2.3.3. However, Referee 1 commented that the description of the methodology is already very detailed, so we will not further develop on the signal rank computation in the manuscript.

McKinney, Wes. 2010. "Data Structures for Statistical Computing in Python." In *Proceedings of the 9th Python in Science Conference*, edited by Stéfan van der Walt and Jarrod Millman, 51–56.

[Figure]

**2.10  Comment # 2.10**

*Page 9, Line 20. "the two signals" should be clearly identified as the SOFI signal and the reference signal.*

Thank you for pointing this out.

**Changes**: We will identify the two signals as suggested.

**2.11  Comment # 2.11**

*Page 12, Figure 6. Only the case of video FloodXCam5 was given, please add cases of parking lot and park that are the typical scenes of urban flooding.*

It is true that these other cases could be of interest to readers. We had left them out for space reasons, but they can easily be added.

**Changes**: We will add the two additional cases to Figure 6.

**2.12  Comment # 2.12**

*Page 14, Line 2. "the visually estimated flooding intensity" and "Flooding intensity" in Figure 8 should be consistent with "the visually estimated water level" in the caption of Figure 8.*

Thank you for pointing this out.

**Changes**: We will change the figure caption accordingly.

2.13  Comment # 2.13

*In the "Discussion" section, more comparisons with other existing methods for urban flooding information extraction from surveillance images should be added.*

Thank you for this comment. We were originally not sure how much to develop comparisons, but are happy to do add a few more.

**Changes**: We will add additional comparisons to existing methods for urban flood information extraction from surveillance images, showing both advantages and disadvantages of the alternatives. The publication of Jiang et al. (2019) proposed in comment # 2.2 will be included.

2.14  Comment # 2.14

*The accuracy of this approach could be affected by camera inclination and manual labeling. Topography can also affect the accuracy, e.g., in flat areas, the visible area of the flooding varies greatly, but the change of water depth value may be very small, while in low-lying areas, the change of visible area of the flooding is small, but the change of water depth value may be very big. These error sources should be discussed in the "Discussion" section.*
Thank you for this comment. It is true that a variety of factors can affect the accuracy of our approach. Although some error sources are already mentioned in the Discussion, it is true that we could have developed this further.

**Changes**: we will add an additional discussion of influencing factors after Section 4.1.

2.15   Comment # 2.15

*In the "Conclusions" section, the authors should explain the direct applications and managerial implications of this approach.*

Thank you for this comment. We agree that a broader perspective was lacking in the Conclusions section.

**Changes**: In the Conclusions, we will relate how our method could be applied in practice and how it could change the management of urban water systems.

---

## Short Comment (SC1) · 10 Apr 2019

The link to the Flood-PREPARED project in our reply to comment #1.3 was lost during formatting. A description of the project can be found here: https://gtr.ukri.org/projects?ref=NE%2FP017134%2F1

---

## Author Response (AR1)

**Referee #1: Comments, authors' replies, and changes**

**General reply to Referee #1's comments**

We thank Anonymous Referee #1 for the positive appreciation of the manuscript and valuable suggestions.

**Comment #1.1**

*Given the target audience will primarily be hydrologists, I find the description of the methodology (particularly the software development and computational methodology) to be very complex and in-depth. I'm sure for some in the community this will be fairly easy language to interact with, but certainly to someone that doesn't design, code or work the tools such as this, I found much of the section passed straight over my head.*

Thank you for this notice. We are aware that this manuscript and the description is outside the discipline of pure hydrology. After considering moving the methodological details to the appendix, we are inclined to leave them in the main body of the article. The reason we find this appropriate is that the methodology presented forms the core novelty of the paper.

**Comment #1.2**

*The presentation of some of the figures are a little poor - while the matrix of images used to highlight how the model has been trained are undoubtedly a worthwhile addition to the paper, I definitely found them a little illegible due to their size - even zooming right in to the PDF rendered some of the images unclear as to what was going on. Furthermore, I originally elected to print a copy of the paper to review while travelling and the images do not translate to black and white or greyscale, perhaps highlighting an accessibility issue.*

Thank you for raising this point.

**Changes**: We changed Figure 4 to improve its readability: we moved the three less interesting case studies to the appendix, so that the size of the images can be increased. Also, we increased the image resolution to 600 dpi. Finally, we modified the colors so that the segmentation boundaries are clear.

[Figure]

Figure 4: Sample frames taken from three of the six analyzed surveillance videos, shown with the human label or automatic flood segmentation in blue and the regions of interest (ROI) in red. The samples show how the "Augmented" strategy improves segmentation in the *Garage* and *Park* samples, but the segmentation for *FloodXCam1* is only successful with the "Fine-tuned" training strategy. The sample frames from the remaining three videos are provided in the supplementary materials.

Moy de Vitry, Matthew  March 19, 2019

Moy de Vitry, Matthew  March 19, 2019

Moy de Vitry, Matthew  March 19, 2019

**Comment #1.3**

*Appreciating the author's comments about copyright and third party ownership of CCTV imagery, I do feel a specific case study may have aided the paper rather than just the catalogue of training videos presented in section 3.1. I understand this may not be an option for the authors, based on the nature of the copyright imagery they had access to, but it would have been useful to see an example using real data and images from an observed urban flood event.*

It is true that a specific case study of a real urban flood event would be beneficial to show the practical feasibility of the method proposed. Unfortunately, the possibility to collect such data would require a long-term study that was not within the means of the project. Nevertheless, other projects that aim to collect real flooding data could pick up this point. For example, the data collected in the ongoing Flood-Prepared project (link) might be a good candidate for such an applied study.

**Changes**: We mentioned the need for a more in depth, practical case study of urban flooding in the discussion section of the manuscript.

**4.7 Recommendations for future research**

Future research should assess the actual value of the information provided by SOFI for the validation and calibration of urban flood models. For this, it would be of value to have a large-scale, long-term case study of a flood-prone urban area, in which surveillance footage and reference flood measurements are available. Additionally, possible methods to de-noise the SOFI signal and quantify its reliability should be investigated.

**Comment #1.4**

*Finally, I found the discussion section to be a little too focused on the technical aspects and challenges of the methodology. What would have been nice, was a slightly more "big picture" discussion that highlight the impacts and possible extensions of this work in an urban setting. Some of the findings here would be significant for urban drainage managers, infrastructure managers (particularly transport etc), quantifying economic risks associated via damage etc and future policy creation around climate change scenarios and drainage dimensions. I acknowledge the authors touched on some of these issues in the introductory section, but it would have been a nice conclusion to the paper*

*to discuss how some of their results may help remove some of the challenges that they identified in the introduction.*

Thank you for this suggestion.

**Changes**: In the Conclusions section of the manuscript, we have elaborated on the possibilities that the proposed method opens up, at the same "big-picture" level as the introduction.
* * *
20 **5 Conclusions**

In this study, we explored the potential of using a deep convolutional neural network (DCNN) and a simple but novel index (SOFI) to obtain flood level trend information from generic surveillance cameras. The results of our study strongly suggest that qualitative flood level information can indeed be extracted automatically and universally from any static camera, although we see the need for many training images to cover the range of appearances that floodwater can take. To

25 compensate for the limited number of training images available in this study, we found that degrading image quality during training improved segmentation performance by approximately 10% (IoU) on low-quality surveillance images. Additionally, fine-tuning the DCNN to a specific video with as few as seven manually labeled images further improved performance. In our results, the SOFI signal from the camera-independent DCNN correlated with water level trends at a rate of 75% on average (Spearman rank correlation coefficient). The proof of concept presented in this study has significant implications.

30 Namely, it confirms that footage from surveillance cameras, processed with artificial intelligence, could provide urban flood

monitoring data for entire cities while preserving privacy. Without the need for specialized instrumentation or field measurements, monitoring could be conducted at a relatively low cost, which is especially attractive for cities with limited financial resources. Previous research suggests that the qualitative information contained in SOFI can easily be used to reduce parameter uncertainty in urban flood models. Apart from modelling-related applications, SOFI could also be useful

5 for coordinating flood response and verifying insurance claims.
* * *
**Referee #2: Comments, authors' replies, and changes**

**Reply to general comments from Referee # 2**

*This paper proposed an approach to monitor flood level trend using DCNN. The topic is very interesting. However, in my opinion, it could be difficult for modelers, decisionmakers and city planners to use the Static Observer Flooding Index (SOFI) directly. The authors should clearly explain what the direct or specific application scenarios of SOFI are. If SOFI or visible area of the flooding can be converted into water depth value or even class information on water depth, it would make this approach more useful and valuable. Unfortunately, this paper lacks this attempt, so I suggest rejecting this paper but encourage resubmission after the improvements have been made.*

We thank Referee 2 for the overall positive review, for the useful recommendations, and for the questions, which have allowed us to further improve the manuscript.

**On the value of flood level trend information**

While we agree with Referee 2 that decision-makers and city planners may not be accustomed to flood trend information, modelers in research have already successfully used trend data for model calibration (e.g. van Meerveld et al 2017), albeit in the field of natural catchment hydrology. In that publication, stream level class data are used as trend information in a conventional calibration scheme, with the minor adaptation being the use of the Spearman rank correlation coefficient as an objective function. Similarly, Wani et al. (2017) showed that even simple binary information of a combined sewer overflow in an urban catchment can reduce model parameter uncertainty. Making the link to our manuscript, binary flooding information can easily be derived from the flood trend information that SOFI provides, by defining for each camera a baseline SOFI value below which no flooding is assumed. We agree with Referee 2 that our manuscript could better explain how the utility of SOFI is supported by literature. To do so, we have augmented our manuscript with details and background information for the examples listed above. We made these changes in the introduction, discussion, and conclusion of the manuscript (see "changes" below).

**On the advantages of SOFI over water depth estimation**

Referee 2 also suggests converting SOFI into water depths (either values or classes) if possible. Although methodologies for determining flood water levels are of value, we argue that the concept of flood trend monitoring with SOFI has advantageous characteristics that would be lost if SOFI were to be converted into a water depth.

1. Higher scalability.
   The obtention of water depth information from a segmented flooding image requires knowledge of the dimensions of reference objects in the camera scene. As shown by the literature suggested by Referee 2, there are different ways of obtaining this information. For example, one can rely on the visibility of ubiquitous objects of known dimensions (Jiang et al. 2019). This approach is limited to scenes in which such objects are visible, and fails if the reference object is hidden or partially hidden by mobile objects during a flood. Another way of obtaining reference measurements is with a survey of a large feature in each scene. Bohla et al. (2018) demonstrate this method nicely with bridges in urban stream settings. Needless to say, this method requires

time, effort, and field knowledge for each of the sites surveyed, making it expensive and time-consuming to deploy at scale. SOFI was conceived as a metric that has few prerequisites and relies on minimal on-site knowledge, making it applicable to a broader range of scenes than any alternative method known to the authors.

2.  Fewer errors.
    In stream or river settings, direct conversion of SOFI to a water depth is most probably possible e.g., by determining a "SOFI-water depth" curve analogous to an H-Q curve. However, in an urban environment, the movement of objects (e.g., cars, people) constantly change the extent of visible water. It follows that the "SOFI-water depth" curve will be inherently wrong. By setting lower ambitions, SOFI also relies on fewer assumptions about the scene monitored, and thereby introduces fewer possible sources of error.

3.  Lower risk of overvaluation.
    Although a conversion of SOFI into water depth (either absolute value or class) could make the data more interpretable, it would dissimulate uncertainty about the information delivered and could risk being valued at the same level of trust as data from conventional sensors. We therefore see value in keeping SOFI an adimensional, qualitative metric.

**Summary**

In summary, literature supports the argument that the flood level trend information reflected in SOFI is valuable in itself to improve urban flood analysis, and do not necessarily need to be converted into water depth. Furthermore, the conversion of SOFI into a water depth would not only reduce the ease of applicability, but also introduce additional error sources and lead to possible overvaluation. Based on these arguments, we do not see the need to transform SOFI into a water level estimation method.

**Changes**: However, we understand that the value of SOFI can be clarified and we have done this in Sections 1.1, 4.6, and 5.

**1 Introduction**

20    **1.1 The need for urban pluvial flood monitoring data**

[revised manuscript text omitted]

**5 Conclusions**

In this study, we explored the potential of using a deep convolutional neural network (DCNN) and a simple but novel index (SOFI) to obtain flood level trend information from generic surveillance cameras. The results of our study strongly suggest that qualitative flood level information can indeed be extracted automatically and universally from any static camera, although we see the need for many training images to cover the range of appearances that floodwater can take. To compensate for the limited number of training images available in this study, we found that degrading image quality during training improved segmentation performance by approximately 10% (IoU) on low-quality surveillance images. Additionally, fine-tuning the DCNN to a specific video with as few as seven manually labeled images further improved performance. In our results, the SOFI signal from the camera-independent DCNN correlated with water level trends at a rate of 75% on average (Spearman rank correlation coefficient). The proof of concept presented in this study has significant implications. Namely, it confirms that footage from surveillance cameras, processed with artificial intelligence, could provide urban flood monitoring data for entire cities while preserving privacy. Without the need for specialized instrumentation or field measurements, monitoring could be conducted at a relatively low cost, which is especially attractive for cities with limited financial resources. Previous research suggests that the qualitative information contained in SOFI can easily be used to reduce parameter uncertainty in urban flood models. Apart from modelling-related applications, SOFI could also be useful for coordinating flood response and verifying insurance claims.

We now also present the advantageous characteristics of SOFI in an extended comparison with alternative methods, as requested by Referee 2 in Comment # 2.13 (see comment #2.13 for changes).

**References**

Bhola, Punit Kumar, Bhavana B. Nair, Jorge Leandro, Sethuraman N. Rao, and Markus Disse. 2018. "Flood Inundation Forecasts Using Validation Data Generated with the Assistance of Computer Vision." *Journal of Hydroinformatics* 21 (2): 240–56. https://doi.org/10.2166/hydro.2018.044.

Jiang, Jingchao, Junzhi Liu, Changxiu Cheng, Jingzhou Huang, and Anke Xue. 2019. "Automatic Estimation of Urban Waterlogging Depths from Video Images Based on Ubiquitous Reference Objects." *Remote Sensing* 11 (5): 587. https://doi.org/10.3390/rs11050587.

van Meerveld, H. J. I., Vis, M. J. P., and Seibert, J.: Information content of stream level class data for hydrological model calibration, Hydrol. Earth Syst. Sci., 21, 4895-4905, https://doi.org/10.5194/hess-21- 4895-2017 , 2017.

*Wani, Omar, Andreas Scheidegger, Juan Pablo Carbajal, Jörg Rieckermann, and Frank Blumensaat. 2017. "Parameter Estimation of Hydrologic Models Using a Likelihood Function for Censored and Binary Observations." Water Research 121 (September): 290–301. https://doi.org/10.1016/j.watres.2017.05.038*

**Comment # 2.1**

*Page 1, Line 16 "The results suggest that the approach can be used with almost any surveillance footage ". I think this conclusion may be too strong.*

Thank you for pointing this out. This conclusion is indeed too strong given the limited number of videos on which the method was demonstrated (even though we chose videos that are diverse from a quality and scenery standpoint).

**Changes**: We now instead write that the results confirm that the method is versatile and can be applied to a variety of camera models and flooding situations.

**Abstract.** In many countries, urban flooding due to local, intense rainfall is expected to become more frequent because of climate change and urbanization. Cities trying to adapt to this growing risk are challenged by a chronic lack of surface flooding data that is needed for flood risk assessment and planning. In this work, we propose a new approach that exploits existing surveillance camera systems to provide qualitative flood level trend information at scale. The approach uses a deep convolutional neural network (DCNN) to detect floodwater in surveillance footage and a novel qualitative flood index (SOFI) as a proxy for water level fluctuations visible from a surveillance camera's viewpoint. To demonstrate the approach, we trained the DCNN on 1281 flooding images collected from the Internet and applied it to six surveillance videos representing different flooding and lighting conditions. The SOFI signal obtained from the videos had on average a 75% correlation to the actual water level fluctuation. By retraining the DCNN with a few frames from a given video, performance for that video is further increased to 85% on average. The results confirm that the approach is versatile, with the potential to be applied to a variety of surveillance camera models and flooding situations without the need for on-site camera calibration. Thanks to this flexibility, this approach could be a cheap and highly scalable alternative to conventional sensing methods.

**Comment # 2.2**

*Some new related references about automatic water level monitoring with surveillance images should be included to strengthen this paper. Wang, R. Q., Mao, H., Wang, Y., Rae, C., & Shaw, W. (2018). Hyper-resolution monitoring of urban flooding with social media and crowdsourcing data. Computers & Geosciences, 111, 139-147. Jiang, J., Liu, J., Cheng, C., Huang, J., & Xue, A. (2019). Automatic Estimation of Urban Waterlogging Depths from Video Images Based on Ubiquitous Reference Objects. Remote Sensing, 11(5), 587. Bhola, P. K., Nair, B. B., Leandro, J., Rao, S. N., & Disse, M. (2019). Flood inundation forecasts using validation data generated with the assistance of computer vision. Journal of Hydroinformatics, 21(2), 240-256.*

Thank you for referring us to these recent publications, which both highlight the need for flood monitoring data and illustrate the complexity of extracting absolute water level information from images.

**Changes**: We have added the reference to Wang et al (2018) to Section 1.1.

Abderrezzak et al., 2009; Leandro et al., 2009). In this context, researchers and practitioners have turned to alternative sources of data such as surveillance footage (Liu et al., 2015; Lv et al., 2018), ultrasonic-infrared sensor combinations (Mousa et al., 2016), field surveys (Kim et al., 2014), social media and apps (Wang et al., 2018), and first-hand reports (Kim et al., 2014; Yu et al., 2016).

And the other two references have been added to Section 1.3

**1.3 Automatic water level monitoring with surveillance images**

While manual reading of water level from surveillance images is possible (e.g. in the study of Liu et al. (2015)), it is both
5    prohibitively labor-intensive at scale and potentially critical from a privacy perspective. Automatic image analysis helps overcome these hurdles, and has already been the subject of research. The following publications provide the current state of the art of automatic water level estimation from ground-level images.

In the work of Lo et al. (2015), video frames are segmented into a number of visually distinct areas using a graph-based approach. The area corresponding to water is identified thanks to an operator-provided "seed", and the water level is
10    qualitatively assessed by comparing the water area to virtual markers placed in the image by the operator. With a more camera-specific solution, Sakaino (2016) estimates water levels with a supervised histogram-based approach which assumes a straight water line on a wall visible in the footage. Similarly, Kim et al. (2011) used a ruler in the camera's field of view as a reference for the water level measurement. A similar approach is used by Bhola et al. (2018), who used the size of large objects like bridges to estimate the real height of automatically detected water surface in the image. Although these methods
15    work well, they rely on in-situ measurements and site-specific calibration, and may be challenging to apply to a large number of cameras.

A more modern approach for image-based flood level estimation has been proposed by Jiang et al. (2018a). The authors use a deep convolutional neural network to extract image features and then apply a regression to infer water level. Although the results are positive, the approach requires that the neural network and regression be retrained for each camera. Thus, the
20    method is probably most valuable for providing redundancy to existing water level readings and not as a scalable flood monitoring solution. Recently, a new approach has been proposed, which theoretically overcomes this problem by estimating water depth from the immersion of ubiquitous reference objects (e.g. bicycles) of known height (Chaudhary et al., 2019; Jiang et al., 2019). However, this approach requires that such objects be visible in the scene in order to provide information.

**Comment # 2.3**

*Page 4, Figure 1. There should be a "surveillance images" box between "camera" box and "deep conv. network" box.*

Thank you for this input. We had several versions of Figure 1, some of which contained the box proposed, but left it out in the final version.

**Changes**: We have introduced a "video frames" box between "camera" box and "deep conv. neural network" box.

[Figure]

**Figure 1:** As an alternative to conventional sensors (top), flood trend information is extracted from surveillance footage by computing the fractional water-covered area (SOFI) over a series of video frames (bottom).

**Comment # 2.4**

*Page 4, Line 18. Why U-net was selected for water segmentation?*

U-Net is a very well known DCNN architecture for semantic segmentation, as the method has nearly 5000 citations in Google Scholar. It is well-suited to the flooding segmentation problem because of its relatively compact size compared to more recent state-of-the-art architectures (such as Mask-RCNN). The smaller size makes it both easier to train with small datasets (which we have) and faster to run, which is useful for flood monitoring.

**Changes**: We have included these reasons in the manuscript.

**2.1.2 Water segmentation with U-net**

The DCNN architecture used for water segmentation in this work is that of U-net (Ronneberger et al., 2015). U-net builds on the FCN architecture, but differs in that the decoding layers have as many features as their respective encoding layers, which allows the network to propagate context and texture information to the final layers. Additionally, U-net implements "skip"

5   connections to preserve details and object boundaries, by carrying information directly from the encoding to the decoding layers. The U-net architecture is well-suited to the water segmentation problem because of its relatively compact size compared to more recent state-of-the-art semantic segmentation architectures, such as Mask R-CNN (He et al., 2017). The smaller size makes it both easier to train with small datasets (like the one available for this study) and faster to run, which is useful for flood monitoring. To code the DCNN, we built on an open source implementation of U-net (Pröve, 2017) that uses

10   Keras (Chollet and others, 2015) to interface with the TensorFlow library (Abadi et al., 2016).

**Comment # 2.5**

*Page 6, Lines 18-20. In the augmented strategy, how many images were used to train the U-net model? Is the augmented data the same amount as the basic data? Moreover, Page 7, Table 1, the augmentation steps in the augmented strategy should be clearly explained.*

Thank you for this question. For the augmented strategy, we use the same images as for the basic strategy. Like for the basic strategy, each training image is fed into the network up to 200 times during training (fewer if the training is completed faster). For the augmented strategy, however, each image is first randomly transformed before being fed into the DCNN.

**Changes**: We have added information about the augmented strategy in Section 2.1.3. Furthermore, we have provided more details about the augmentation transformations applied.

> In the **Augmented** strategy, the same images as for the basic training strategy were used but with additional augmentation
> steps that degraded image quality to the level of typical surveillance footage. The following image transformations,
> 10    implemented with the Keras library (Chollet and others, 2015) were applied during augmentation:
>
> - random horizontal mirroring
> - translate image horizontally and vertically by +/- 20%
> - change in contrast by +/- 50%
> - resolution deterioration by zooming into image at different locations up to 33%
> 15 - decreasing saturation by up to 80%
> - alter image brightness between -80 and +20 units
> - blurring with random Gaussian filter
>
> Augmentation was applied with a 20% probability each time an image was fed into the network for training (up to 200 times,
> corresponding to the number of epochs).

**Comment # 2.6**

*Page 7, Lines 5-6. It is not accurate enough. How many seconds or minutes does each model training take?*

**Changes**: At the line indicated, instead of a range we now indicate the approximate average training time in minutes for the basic and augmented strategies separately.

> To train the network, the Adaptive Moment Estimation (Adam) was chosen as the gradient descent optimizer because it
> shows good convergence properties (Kingma and Ba, 2015). The dice coefficient served as the loss function, defined after
> 5    Zou et al. (2004). The DCNN was trained on an Nvidia Titan X (Pascal) 12 GB GPU. The "Basic" strategy took
> approximately 120 minutes on average, whereas the "Augmented" strategy required approximately 180 minutes for training.
> The fine-tuning process required an additional 5 minutes of training per video.

**Comment # 2.7**

*Page 8, Table2. The total frames or minutes and the resolutions of surveillance images should be given. How to define the quality of surveillance footage should be explained.*

It is true we could have provided more information here about the videos and the quality terms used.

**Changes**: We added the requested information to Table 2. "Quality" has been replaced with "visual clarity" and is specified for each video. (The tables have been moved to the end of the manuscript to match the manuscript preparation guidelines).

TABLE 2: SURVEILLANCE FOOTAGE USED IN STUDY

| Video | Setting | Flood type | Image resolution | Water level trend | Analysis frequency (frames/min) | Event duration | Visual clarity of flooding | Source |
|-------|---------|-----------|------------------|-------------------|-------------------------------|----------------|---------------------------|--------|
| FloodX Cam1 | Experimental facility, weir | Artificial | 1280x720 | Sensor | 60 | 164 minutes | High: Wet surfaces | (Moy de Vitry et al., 2017) |
| FloodX Cam5 | Experimental facility, cellar | Artificial | 1280x720 | Sensor | 60 | 166 minutes | Medium: Wet surfaces, desaturated image | (Moy de Vitry et al., 2017) |
| Garage | Indoor, garage | Extreme rainfall | 1280x720 | Visual | 24 | 19.8 hours | High: Moving objects floating in water | (Roisman, 2017) |
| Park | Outside, sport field | River flooding | 640x480 | Visual | 1 | 13.5 hours | Low: Camera behind window with reflections; camera is sometimes | (Cityofchaska, 2010) |

**Comment # 2.8**

*Please add the average segmentation time of a single frame for the testing as a performance, since the real-time monitoring is usually important.*

Thank you for this request; it is true that this figure could be of interest to readers.

**Changes**: We have provided average segmentation time in Section 3.1

**3 Results**

15  **3.1 Automatic flood water segmentation**

Image segmentation, with the setup described in Section 2.1.3, takes around 50ms per image. Figure 4 provides sample frames from three of the six surveillance videos used in this work. The other three videos are provided in Figure 1 of the supplementary material. In each case, the human labels as well as automatic segmentations from the various DCNNs are drawn in blue. Additionally, the ROIs are drawn in red for each video, defined manually according to the criteria mentioned

20  in section 2.2.

**Comment # 2.9**

*Page 9, Lines 16-17. The authors should explain more extensively how to compute the rank of each signal value.*

Thank you for your interest. The computation of the Spearman rank-order correlation coefficient, including signal rank of each value, was performed using the pandas Python library (McKinney, 2010), which is open source.

**Changes**: We have added a reference to this library in Section 2.3.3. However, Referee 1 commented that the description of the methodology is already very detailed, so we did not further develop on the signal rank computation in the manuscript.

McKinney, Wes. 2010. "Data Structures for Statistical Computing in Python." In *Proceedings of the 9th Python in Science Conference*, edited by Stéfan van der Walt and Jarrod Millman, 51–56.

> an in-situ sensor or by visual inspection of the surveillance footage, as described in Section 2.3.1. The pandas Python library (McKinney, 2010), which contains an implementation of the Spearman rank correlation coefficient, was used for time series analysis.

**Comment # 2.10**

*Page 9, Line 20. "the two signals" should be clearly identified as the SOFI signal and the reference signal.*

Thank you for pointing this out.

**Changes**: We have identified the two signals as suggested.

> coefficient, the rank of each signal value must be computed relative to its respective signal. For signals in which the same value can appear multiple times (tied ranks), the Spearman rank-order correlation coefficient $\rho$ is given by
>
> $$\rho = \frac{\Sigma(x_i - \bar{x})(y_i - \bar{y})}{\sqrt{\Sigma(x_i - \bar{x})^2 \Sigma(y_i - \bar{y})^2}}. \qquad (3)$$
>
> where $x_i$ and $y_i$ are the ranks of the two signals for time step $i$, and where $\bar{x}$ and $\bar{y}$ are the average ranks of the SOFI and water level signals, respectively. In the current study, the reference signal for the water level trend was obtained either from

**Comment # 2.11**

*Page 12, Figure 6. Only the case of video FloodXCam5 was given, please add cases of parking lot and park that are the typical scenes of urban flooding.*

It is true that these other cases could be of interest to readers. We had left them out for space reasons.

**Changes**: We have added all other video cases to the supplementary materials.

> Figure 6: Water level (blue) and SOFI signal for whole image (black) or Region of Interest (ROI, red) for video *FloodXCam5*. With more advanced DCNN training strategies, noise and outliers in SOFI are visibly reduced in the ROI. The time series for the remaining videos can be found in the supplementary materials, Figures A2-A6.

**Comment # 2.12**

*Page 14, Line 2. "the visually estimated flooding intensity" and "Flooding intensity" in Figure 8 should be consistent with "the visually estimated water level" in the caption of Figure 8.*

Thank you for pointing this out.

**Changes**: We have changed the figure caption accordingly.

> Figure 8: Videos without reference measurements: SOFI computed from full images (black) or from ROI (red) plotted against visually estimated flood intensity, for different training strategies. There is a visible reduction of noise and increased correlation with the "Augmented" and "Fine-tuned" strategies.

**Comment # 2.13**

*In the "Discussion" section, more comparisons with other existing methods for urban flooding information extraction from surveillance images should be added.*

Thank you for this comment. We were originally not sure how much to develop comparisons, but are happy to do add a few more.

**Changes**: We have added additional comparisons to existing methods for urban flood information extraction from surveillance images, showing both advantages and disadvantages of the alternatives. The publication of Jiang et al. (2019) proposed in comment # 2.2 was referred to.

**4.1 SOFI as a scalable and robust approach for qualitative water level sensing**

Compared to alternative methods for water level monitoring, our approach is less ambitious in the type of information it aims to provide, as it only attempts to communicate water level fluctuation and not an absolute water level. This weakness is at least partially compensated for by an increased scope of applicability, as it is almost the only surveillance image-based monitoring approach designed to provide water level information without needing to be calibrated to each camera. As described in Section 1.3, research has typically been focused on extracting absolute water levels from images, mainly by identifying the water edge in relation to an object that has been measured and manually identified in the image (Bhola et al., 2018; Kim et al., 2011; Sakaino, 2016). Although Lo et al. (2015) present a method that does not necessarily need reference measurements, human intervention is still needed for each camera to define virtual markers and a "seed" to inform water segmentation. The concept presented by Jiang et al. (2019) is an exception, theoretically overcoming the need for per-image calibration by taking ubiquitous objects as scale references. Nonetheless, that method assumes that such reference objects are visible in the flooding scene, which may not always hold. In comparison, SOFI does not need specific objects to be present, nor does it need field measurements or manual set-up of the image. The DCNN-based water segmentation approach used is flexible enough that given sufficient training data, it can theoretically be applied to any situation. In this study, the scalability of SOFI was demonstrated by applying the "Augmented" DCNN to six videos not seen during training, from which information was extracted despite complex environments, moving objects, and bad lighting conditions.

The simplicity of the SOFI approach also gives it robustness. By using areal integration to quantify flood intensity, the index is less sensitive to small segmentation errors, which could be problematic if punctual virtual markers are used as in the case of Lo et al. (2015). Also, since SOFI only claims to provide trend information, systematic segmentation errors (e.g. misclassification at the water-background boundary) should only be of minor consequence. This aspect of SOFI is well illustrated in this study by the *Park* video, for which the SOFI signal had a high correlation to the water level trend despite mediocre segmentation performance.

Besides the advantages of SOFI, its use of a DCNN carries a disadvantage due to its need for training data. Methods that use heuristics to segment water, such as Lo et al. (2015), do not require as much training data, although their application is then often limited to visually similar situations.

**Comment # 2.14**

*The accuracy of this approach could be affected by camera inclination and manual labeling. Topography can also affect the accuracy, e.g., in flat areas, the visible area of the flooding varies greatly, but the change of water depth value may be very small, while in low-lying areas, the change of visible area of the flooding is small, but the change of water depth value may be very big. These error sources should be discussed in the "Discussion" section.*

Thank you for this comment. It is true that a variety of factors can affect the accuracy of our approach. Although some error sources are already mentioned in the Discussion, it is true that we could have developed this further.

**Changes**: We have added an additional discussion of influencing factors after Section 4.1.

**4.2 Factors influencing the information quality of SOFI**

Although SOFI is theoretically applicable at locations where the extent of visible water correlates with flooding depth, there

20 are certain conditions that can negatively influence the quality of information obtained.

First, floodwater can look very different from case to case. For example, (i) water surface movement can cause a range of different wave structures, and the color of water itself is variable, (ii) water reflects light that falls on it, even in a mirror-like fashion if the water surface is still, and (iii) surveillance cameras tend to have low color fidelity, dynamic range, resolution, and sharpness. Due to the complexity of segmenting floodwater, one can expect that more training images would be required

25 than for a typical segmentation problem. In our results, the high inter-video variability in segmentation performance suggests that the number of training images should be increased in future studies. In particular, the variability suggests that the training images were not fully representative of the testing images. Thus, it is probable that the segmentation performance could be substantially improved if a larger and more diverse training set were available.

Second, in situations where large quantities of manually labeled images are required, it is inevitable that some labelling

30 errors occur. Looking into the implications of such errors, Heller et al. (2018) showed that U-net is relatively insensitive to jagged label boundaries. Regarding non boundary-localized errors, all DCNN architectures investigated were found to be very robust.

Finally, the basic assumption that the extent of visible water correlates with flooding depth may not fully hold for every camera scene. For example, the lowest flooding depths may not be visible in the image, and if the camera is oriented such that the whole image can be covered by water, the highest stages of flooding may be censored from the signal as well. These situations will require special handling when the data is assimilated for model calibration. The scene topography and camera

5 placement also affect the slope and linearity of the SOFI-water depth relationship, and can make the trend more difficult to determine if the correlation in the relationship is small. Additionally, the entry and exit of objects in a flooding scene can compromise the approach, especially if the object are large and occur with low frequency in a non-systematic way.

**Comment # 2.15**

*In the "Conclusions" section, the authors should explain the direct applications and managerial implications of this approach.*

Thank you for this comment. We agree that a broader perspective was lacking in the Conclusions section.

**Changes**: In the Conclusions, we now relate how our method could be applied in practice and how it could change the management of urban water systems.

[revised manuscript text omitted]

---

## Referee Report (RR1)

**Summary of Research**

This paper demonstrates the novel application of deep convolutional neural networks to determine the presence of flooding in CCTV footage. In addition, the extent of the flooding can be roughly determined using a new SOFI index. The paper's key findings include:

- The successful segmentation of CCTV images to identify areas covered by water.
- The development of the SOFI index to quantify flooding extent and analyse water level fluctuations.
- SOFI loosely correlates with water depth, which may prove useful to the future calibration of flood models.

The paper demonstrates its results over six CCTV sequences, taken from a variety of locations and test sites. Two examples are also accompanied by water level recordings, providing an objective comparison for the technology.

**Context within current research**

The presented work appears to be new and novel, contributing to a small but growing pool of work on the subject. Other work in the field has concentrated on detecting flood depth from CCTV images or application to still images (particularly from social media). Although the application of deep learning (convolutional neural networks) is widespread across computer vision problems, this is a novel application, and the technology's implementation has been thoroughly explained in this paper.

**Strengths and weaknesses**

Overall, I feel the research presented in this paper is of a high quality, and provides a plethora of insights into the technology and its potential applications. The results are presented in a clear manner, which should be accessible to all readers. However, I was a little surprised by the choice of journal for publication. The paper doesn't feel like it fits perfectly with the journal's target audience, even though the work presented within is of extremely high quality. I tend to agree with the previous reviews, that the technical elements of the methodology may be hard to follow for readers not familiar with deep learning. Even so, the technical content is well referenced, enabling a reader to further explore and understand the more complex elements.

Other reviews have questioned the usefulness of the SOFI descriptor and the conversion to water depths. I tend to agree with the author that SOFI works well as a distinct tool. This does limit its usefulness for existing flood modelling, but it can still be used for contextual validation, even if that is only in a binary manner (flooding present or not). The translation water depths from CCTV footage is a very different and extremely complex problem, given the tremendous volume of noise in 'wild' CCTV footage.

**Comments**

**Key points**

Page 7 Line 21: You describe the 'Fine-Tuning' process, however you don't comment on the viability of this for mass implementation. This could be particularly problematic as you have moved from a single holistic DCNN to many (independent) machines. Furthermore, this does imply that you have footage containing a flood for that camera feed, which is extremely unlikely, especially if someone planned to roll this out to tens of thousands of CCTV cameras. Generally speaking, I wouldn't rely on this fine-tuning process and believe it to be extremely situational in its usefulness.

Page 8 Line 27: Following on from the scalability issues with 'Fine-Tuning', the manual definition of ROIs would not be viable for mass implementation. Even though you found the use of ROIs to be unnecessary, it may be worth the investigation of automatic ROI generation (in future work). Not only would this improve the scalability of the technique, but could improve the calculation of a SOFI index in video containing water multiple water sources.

**Other notes**

From the case studies provided, the technology appears to have been demonstrated on largely still/slow moving water. It would be good if you could comment on the techniques application to moving water (particularly relevant in flash flooding) as still/white water are visually quite distinct.

Another potential application for this technology may be in key infrastructure/assets (i.e generators/pumping stations/power stations) that are particularly at risk of damage to flooding. Quite often these assets will have CCTV cameras installed for security. This technology could act as an additional alarm/early warning system for these at-risk assets and asset failures.

Page 8 Line 19: You discuss issues arriving with your SOFI descriptor if the scene changes suddenly (something is moved by flood water or a vehicle parks in the scene). However, this should be visible in the SOFI curve? Assuming you could work with dynamic thresholds/filtering, this issue could be could be tackled in future work.

Page 8 Line 30: Sentence starts 'provides characteristics of these videos' I assume a word is missing? Otherwise I would advise re-wording.

---

## Author Response (AR2)

**Referee 1 Comments - Author Reply**

**General reply from authors**: We thank Referee 1 for taking the time to review the manuscript again and for the additional comments that helped further improve the manuscript quality.

**General comments**

**Urban flooding usually causes traffic jams. When a large number of vehicles appear in the video image, the visible area of the flooding will become very small, but the flooding depth value may be large. In this situation, the flood level trend will have a large error. I think this should be taken into account in practical applications.**

**Author reply**: For this general comment, we agree that flooding can have a causal relationship to traffic jams and, if vehicles are held up in the water, this will reduce the amount of flooding visible.

The manuscript has been modified to mention this possibility when discussing the factors influencing the data quality of SOFI. Page 18, lines 8-11 (in the tracked-changes document) now read:

Additionally, the entry and exit of objects in a flooding scene can compromise the approach, especially if the objects are large and occur with low frequency. The flooding itself may increase the occurrence of such obstructions, for example in the case of cars stuck in traffic or objects transported by the water.

**Language issues**

Page 17, Line 23: "water reflects light that falls on it, even in a mirror-like fashion if the water surface is still."
The "even" should be replaced with "especially".

Page 18, Line 2: "the lowest flooding depths may not be visible in the image."
The "depths" should be replaced with "areas".

**Author reply:** The first point has been taken and the word "even" has been replaced with "at times".
The second point has also been taken and the sentence has been modified to read (page 18, line 4):

For example, small floods may not be visible due to obstructions, and if the camera is oriented such that the whole image can be covered by water, the highest stages of flooding may be censored from the signal as well.

**Referee 2 Comments - Author Reply**

**General reply from the authors**: We thank referee 2 for taking the time to review the manuscript, and appreciate the positive appraisal of our work. We are grateful for the specific comments and suggestions that have allowed us to further improve the manuscript.

**Key points**

**Page 7 Line 21: You describe the 'Fine-Tuning' process, however you don't comment on the viability of this for mass implementation. This could be particularly problematic as you have moved from a single holistic DCNN to many (independent) machines. Furthermore, this does imply that you have footage containing a flood for that camera feed, which is extremely unlikely, especially if someone planned to roll this out to tens of thousands of CCTV cameras. Generally speaking, I wouldn't rely on this fine-tuning process and believe it to be extremely situational in its usefulness.**

**Author reply**: Regarding the viability of "Fine-Tuning" for mass implementation, the analysis of referee 2 is correct; the process of retraining the DCNN would make the method less scalable. A sentence has been added on page 7, line 26:

However, the additional effort required by this training approach limits its practical utility to situations where the increase in data quality is of particularly high value.

**Page 8 Line 27: Following on from the scalability issues with 'Fine-Tuning', the manual definition of ROIs would not be viable for mass implementation. Even though you found the use of ROIs to be necessary, it may be worth the investigation of automatic ROI generation (in future work). Not only would this improve the scalability of the technique, but could improve the calculation of a SOFI index in video containing water multiple water sources.**

**Author reply**: The automatic generation of ROIs could be a way of reducing error in SOFI. A sentence has been added in the recommendations for future research on page 19, line 25:

Another approach to this problem is through automatic definition of ROIs that censor out noise-generating areas of the video scene.

**Other notes**

**From the case studies provided, the technology appears to have been demonstrated on largely still/slow moving water. It would be good if you could comment on the techniques application to moving water (particularly relevant in flash flooding) as still/white water are visually quite distinct.**

**Author reply**: We have modified the discussion about factors influencing data quality to account for this issue. A point (ii) has been added to the list of factors such as water color and light reflections on page 17, line 23:

… (ii) the color of the water is variable and in very agitated flows, bubbles can make the water appear white,

**Another potential application for this technology may be in key infrastructure/assets (i.e generators/pumping stations/power stations) that are particularly at risk of damage to flooding. Quite often these assets will have CCTV cameras installed for security. This technology could act as an additional alarm/early warning system for these at-risk assets and asset failures.**

**Author reply**: It is true that warning systems are another application of the technology proposed. A sentence mentioning this application has been added to the discussion of the practical value of SOFI on page 19, line 17:

Similarly, flood detection could be used for warning systems of critical infrastructure such as generators, pumps, or power transformers.

**Page 8 Line 19: You discuss issues arriving with your SOFI descriptor if the scene changes suddenly (something is moved by flood water or a vehicle parks in the scene). However, this should be visible in the SOFI curve? Assuming you could work with dynamic thresholds/filtering, this issue could be could be tackled in future work.**

**Author reply**: Yes, filtering may be an option. Nevertheless, thresholds generally involve some kind of parameter tuning (which reduces scalability). A sentence has been added to the recommendations for future research on page 19, line 23:

In particular, the issue of rapid changes in the SOFI signal due to moving obstructions could be addressed by filtering changes that surpass a given threshold.

**Page 8 Line 30: Sentence starts 'provides characteristics of these videos' I assume a word is missing? Otherwise I would advise re-wording.**

**Author reply**: Indeed, a reference to Table 2 was missing. Thank you for pointing this out.

[revised manuscript text omitted]